



# Improved 1-km-resolution PM$_{2.5}$ estimates across China using the space-time extremely randomized trees

Jing Wei[1], Zhanqing Li[2*], Wei Huang[3], Wenhao Xue[1], Lin Sun[4], Jianping Guo[5], Yiran Peng[6], Jing Li[7],

Alexei Lyapustin[8], Lei Liu[9], Hao Wu[1], Yimeng Song[10]

1. State Key Laboratory of Remote Sensing Science, College of Global Change and Earth System Science, Beijing Normal University, Beijing, China
2. Department of Atmospheric and Oceanic Science, Earth System Science Interdisciplinary Center, University
of Maryland, College Park, MD, USA
3. State Key Laboratory of Remote Sensing Science, Faculty of Geographical Science, Beijing Normal University, Beijing, China
4. College of Geomatics, Shandong University of Science and Technology, Qingdao, China
5. State Key Laboratory of Severe Weather, Chinese Academy of Meteorological Sciences, Beijing, China
6. Ministry of Education Key Laboratory for Earth System Modeling, Department of Earth System Science, Tsinghua University, Beijing, China
7. Department of Atmospheric and Oceanic Sciences, School of Physics, Peking University, Beijing, China
8. Laboratory for Atmospheres, NASA Goddard Space Flight Center, Greenbelt, Maryland, USA
9. College of Earth and Environmental Sciences, Lanzhou University, Lanzhou, China
10. Department of Urban Planning and Design, Faculty of Architecture, The University of Hong Kong, Hong Kong

Correspondence to: Zhanqing Li (zli@atmos.umd.edu)

## Abstract

Fine particulate matter with aerodynamic diameters ≤ 2.5 μm (PM$_{2.5}$) shows adverse effects on human health and atmospheric environment. Satellite-derived aerosol products have been intensively adopted in estimating surface PM$_{2.5}$ concentrations, but most previous studies failed to monitor air pollution over

small-scale areas limited by coarse spatial-resolution (3–50 km) and low data-quality aerosol optical depth (AOD) products. Therefore, a new space-time extremely randomized trees (STET) model is





developed that integrates spatiotemporal information to improve PM$_{2.5}$ estimates at both spatial resolution and overall accuracy across China. To this end, the newly released MODIS MAIAC AOD product, meteorological and other auxiliary data are inputs to the STET model. Daily 1-km PM$_{2.5}$ maps

in 2018 across mainland China are produced. The STET model performs well with a high out-of-sample (out-of-station) cross-validation coefficient of 0.89 (0.88), a low root-mean-square error of 10.35 (10.97) μg/m$^3$, a small mean absolute error of 6.71 (7.17) μg/m$^3$, and a small mean relative error of 21.37 % (23.77%), respectively. Particularly, it can well capture the PM$_{2.5}$ concentrations at both regional and individual site scales. In addition, it posed a strong predictive power (e.g., monthly-R$^2$ =

0.80) and can be used to predict the historical PM$_{2.5}$ records. The North China Plain, the Sichuan Basin, and Xinjiang Province always are featured with high PM$_{2.5}$ pollution, especially in winter. The STET model outperforms most models presented in previous related studies. More importantly, our study provides a new approach to obtain high-quality PM$_{2.5}$ estimates, which is important for air pollution studies over urban areas.


## 1. Introduction

Atmospheric particulate matter is a relatively stable suspension system with solid and liquid particulate matter evenly dispersed. Fine particles are those particles in ambient air with aerodynamic diameters no more than 2.5 micrometers (PM$_{2.5}$). Compared to coarser particles, PM$_{2.5}$ are rich in toxic and harmful

substances and can directly enter the respiratory tract and alveoli of humans. Moreover, they have a long residence time and long transmission distance in the atmosphere (Aggarwal and Jain, 2015). Numerous studies have illustrated that high PM$_{2.5}$ concentration adversely affects human health (Peng et al., 2009; Bartell et al., 2013; Chowdhury and Dey, 2016; Crippa et al., 2019; Song et al., 2019), severely impairs the atmospheric environment (Li et al., 2017), and even significantly influences the

cloud and precipitation systems by aerosol radiative and microphysical effects (Koren et al., 2014; Li et al., 2016; Seinfeld et al., 2016; Ceca et al., 2018). Silva et al. (2013) have shown that about 2.1 million people have died each year, resulting from the increasing PM$_{2.5}$ around the world.
Nowadays, air pollution is becoming more severe due to continuously increasing anthropogenic aerosols in developing countries, especially in China (He et al., 2011; Huang et al., 2014; Liu et al.,



2017; Zhai et al., 2019). Fine particulate matters have become the primary pollutant in urban environment, garnering much scrutiny from the public (Han et al., 2014; Sun et al., 2016; Wu et al., 2018). Therefore, China Meteorological Administration began to establish ground $PM_{2.5}$ observation network to monitor the urban air quality as early as 2004 (Guo et al., 2009), followed by a denser network established by the Chinese Ministry of Environmental Protection since 2013. However, station-

based monitoring is largely limited by the instruments and climatic conditions and cannot completely reflect air pollution over large areas. Satellite remote sensing technology has led to a variety of operational aerosol products using mature aerosol retrieval algorithms (Levy et al., 2013; Lyapustin et al., 2018), which allows the $PM_{2.5}$ estimations at large scale due to their unanimously positive relationships (Guo et al., 2017).

Over the years, numerous approaches have been proposed to improve the $PM_{2.5}$-AOD relationship. Physical models typically construct physical relationships between surface particulate matter concentrations and satellite AOD products through altitude and humidity corrections (Zhang and Li, 2015). Statistical regression models, e.g., the multiple linear regression model, the linear mixed-effect model, the two-stage model, the geographically weighted regression (GWR) model, have been widely

used for applications due to their simplicity and versatility (Gupta & Christopher, 2009; Ma et al., 2014; Xiao et al., 2017; Yao et al., 2019). Artificial intelligence models mainly involve the machine learning and deep learning models, e.g., the random forest (RF; Brokamp et al., 2018; Chen et al., 2018; Hu et al., 2017), the extreme gradient boosting model (XGBoost, Chen et al., 2019), the back-propagation and generalized regression neural networks (BRNN and GRNN, Li et al., 2017a).

However, $PM_{2.5}$ is jointly affected by numerous factors, e.g., meteorological conditions, human activities, and topography, showing great spatial and temporal heterogeneities. This makes it difficult for above traditional physical and statistical regression approaches to accurately explain and construct $PM_{2.5}$-AOD relationships, leading to poor $PM_{2.5}$ estimates. Despite stronger data mining ability, most artificial intelligence approaches have been simplistically adopted in $PM_{2.5}$ predictions, neglecting their

crucial spatiotemporal characteristics (Chen et al., 2018, 2019; Hu et al., 2017; Li et al., 2017a; Brokamp et al., 2018; Xue et al., 2019). Furthermore, deep learning is highly dependent on the computer performance and is less computationally efficient. On the other hand, most widely used





aerosol products are generated with low spatial resolutions (3–50 km), and thus are seriously limited for applications over small-scale regions such as urban areas.

Focus on these problems, to address the spatiotemporal heterogeneity and improve PM$_{2.5}$ estimates, a new space-time extremely randomized trees (STET) model is developed using the MODIS MAIAC AOD product at 1-km resolution associated with meteorological, land-use, topographic, and population parameters. Then the space continuous 1-km PM$_{2.5}$ maps at different temporal scales covering mainland China in 2018 are generated. Section 2 describes the data sources and integration. Section 3 introduces

the space-time extremely randomized trees (STET) model, and section 4 presents the validation and comparison of our PM$_{2.5}$ estimates across China. Section 5 gives a summary and conclusion.

## 2.  Data sources

### 2.1 PM$_{2.5}$ ground measurements

In this study, the hourly in-situ PM$_{2.5}$ observations at 1583 monitoring stations (Figure 1) across mainland China from 1, January 2017 to 31, December 2018 are collected, and they are then averaged to obtain the daily PM$_{2.5}$ measurements. The PM$_{2.5}$ observations are measured using the tapered element oscillating microbalance approach method or β-attenuation monitors that have undergone further calibration and strict quality control procedures (Guo et al., 2009).

*[Please insert Figure 1 here]*

### 2.2 MAIAC AOD product

The MAIAC algorithm was developed and applied to generate MODIS aerosol products from darkest to brightest surfaces at a 1-km spatial resolution over land (Lyapustin et al., 2011). On 30 May 2018,

official 1-km-resolution MAIAC aerosol products were released and made freely available to all users. This dataset is produced using the revised MAIAC algorithm with continuous improvements in scale transition using spectral regression coefficients, cloud detection, determination of aerosol models, over-water processing, and general optimization in the global aerosol retrieval process (Lyapustin et al., 2018). MAIAC daily aerosol products from Terra and Aqua satellites are collected in 2018 across





China, and the 550-nm AOD retrievals with high quality assurance (QA$_{CloudMask}$ = Clear and

QA$_{AdjacencyMask}$ = Clear) are used.

### 2.3 Auxiliary data

The auxiliary data mainly includes meteorological, land-cover, surface topographic, and population

data. The meteorological variables are collected from ERA-Interim atmospheric reanalysis products,

including the boundary layer height (BLH), evaporation (EP), temperature (TEM), precipitation (PRE),

relative humidity (RH), surface pressure (SP), wind speed (WS), and wind direction (WD). For

meteorological variables, the observations between 1000 to 1400 local time are averaged to be

consistent with satellite overpass times. The land-cover data include the MODIS land use cover and

NDVI products. The topographic data include the surface elevation, slope, aspect, and relief (Wei et al.,

2019a), are calculated from the SRTM DEM product, and the population derived from VIIRS nighttime

lights data. Table 1 provides detailed information about the data sources.

*[Please insert Table 1 here]*

### 130 2.4 Data integration

Terra and Aqua MAIAC AOD products have different spatial coverages due to frequent clouds and

difference in their respective imaging times. Therefore, both Terra and Aqua MAIAC datasets are

combined and merged through the linear regression approach (Eq. 1) to reduce the systematic

differences and enlarge the spatial coverage. By integrating the two datasets, the spatial coverage is

greatly increased by more than 15% over most areas across China, which can lead to wider spatial-

coverage PM$_{2.5}$ maps. More importantly, the number of valid data samples has significantly increased

by approximately 25–32% after combination than just using Terra or Aqua MAIAC products, which can

improve the model training ability.

$$\begin{cases} \tau_T = k_1 \cdot \tau_A + b_1 \\ \tau_A = k_2 \cdot \tau_T + b_2 \\ \tau_C = \mathrm{mean}(\tau_T, \tau_A) \end{cases} \quad (1)$$

where $\tau_T$, $\tau_A$, and $\tau_C$ denote the Terra, Aqua, and combined AODs.





In addition, due to different spatial resolutions, all the 16 auxiliary variables are uniformly aggregated to a 1-km ($\approx 0.01° \times 0.01°$) spatial resolution using the bilinear interpolation approach. After removing invalid or unrealistic values, there are 167,716 matched PM$_{2.5}$-AOD samples and independent variables are collected for 2018 in China.


### 3. Space-time extremely randomized trees

In this study, a tree-based ensemble learning approach, called the extremely randomized trees (ET), is selected to deal with complex supervised regression issues and to construct robust PM$_{2.5}$-AOD relationships. Compared to other tree-based ensemble approaches (e.g., RF), this model splits nodes by

completely randomly selecting cut-points and uses all the training sample learning sample (instead of the bootstrap approach) to grow trees. Therefore, it is with stronger randomness and can efficiently solves variance problems and mines valuable information (Geurts et al., 2006). There are four key steps during the splitting process with the training dataset $S$:

(a) Split a node ($S$). $K$ attributes ($a_1$, …, $a_K$) are selected from all independent attributes in the local

training subset $S$; and then $K$ splits ($s_1$, …, $s_k$) are drawn;

(b) Pick a random split. A subset $S$ and an attribute $a$ are used as inputs to calculate the maximum ($a_{max}$) and minimum ($a_{min}$) value; then a random cut-point $a_c$ uniformly in ($a_{max}$, $a_{min}$) is drawn; and if $a < a_c$, the split $s_i$ (i = [1, k]) is returned;

(c) Calculate the score. The score for a split $s_i$ in a subset $S$ is measured by Equation 2. If the split $s_i$

satisfy that Score($s^*$, $S$) = max{Score($s_i$, $S$)}, the split $s^*$ is returned.

(d) Stop the spilt. If $|S| < n_{min}$, or all attributes or the output are constant in in subset $S$, then output a Boolean (i.e., TRUE).

$$Score(s_i, S) = \frac{var\{y|S\} - \frac{|S_l|}{S} var\{y|S_l\} - \frac{|S_r|}{S} var\{y|S_r\}}{var\{y|S\}} \quad (2)$$

where $S_l$ and $S_r$ represents two subsets related to the two outcomes of a split ($s$), and var{} represents

the variance of the output $y$ in the training set $S$.





In the splitting process of the ET model for numerical attributes, $K$ and $n_{min}$ are the two main parameters, which represents the number of attributes randomly selected at each node and the minimum sample size for splitting a node (Geurts et al., 2006), respectively. They are used to establish an ensemble model with the full training samples by building numerous extra-trees. Last, the estimations

of these extra-trees are summarized through the arithmetic average in regression problems to obtain the result.

### 3.1 Model development

Specifically, spatiotemporal heterogeneities, i.e., strong spatial autocorrelation and obvious temporal

differences, is the key characteristic of PM$_{2.5}$, presenting great challenges and usually neglected in most regression and artificial intelligence models. Therefore, in this study, a new space-time extremely randomized trees (STET) model, which introduces both the spatial and temporal information, is developed to solve this problem. The spatial (Space) information is represented by the geographical difference between two pixels calculated using the Haversine approach based on their longitude and

latitude information (Eq.3), and the temporal (Time) information is represented by the time difference for a given pixel on different days in a year (Eq.5). These two space-time terms can better distinguish and represent the spatiotemporal autocorrelations of PM$_{2.5}$ between different pixels on different polluted days.

$$P_{S(i,j,t)} = f(Lon_{i,j,t}, Lat_{i,j,t}) = haversin(\triangle \alpha) + \cos(\alpha_1)\cos(\alpha_2)\,haversin(\triangle \beta) \quad (3)$$

$$haversin(\theta) = \sin^2(\theta/2) = [1 - \cos(\theta)]/2 \quad (4)$$

$$P_{T(i,j,t)} = DOY_{i,j,t} \quad (5)$$

where $P_{x(i,j,t)}$ represents a given pixel at location ($i$, $j$) in the year $t$, and DOY represents the day of year; $\alpha_1$ and $\alpha_2$ denote the latitude of two points, and $\triangle \alpha$ and $\triangle \beta$ denote the latitude and longitude difference between two points in space. Therefore, surface measured PM$_{2.5}$ concentrations, MAIAC

AODs, meteorological conditions, land cover, topographic conditions, population, and spatiotemporal information are used as preliminary inputs for the STET model.

### 3.2 Model adjustment



However, due to a large number of independent variables considered, this will lead to the unavoidable
over-fitting issue during the model training process. Therefore, the model need be further adjusted by
selecting more important variables rather than all variables to overcome this issue and improve the
model efficiency. For this purpose, the importance scores of all selected independent variables and
spatiotemporal information to PM$_{2.5}$ estimates for the STET model are calculated in China (Figure 2).
The results suggest that AOD is the most influential variable, contributing ~31% toward daily PM$_{2.5}$
estimates. Time and space terms are the other two important factors, contributing about 9–10%. This
further illustrates the importance of spatial and temporal information on PM$_{2.5}$ estimates. Because there
is little precipitation on most days throughout the year, PRE contributes little to PM estimates, by
contrast, most other meteorological variables contribute more to PM$_{2.5}$ estimates, especially BLH, EP,
and TEM with average importance scores of 9%, 8%, and 7%, respectively. The contributions of
surface conditions (i.e., LUC, relief, aspect, and slope) and NTL to PM$_{2.5}$ estimates are generally less
than 2%. Therefore, these six less important variables are excluded from the STET model and the
remaining variables are used to construct the finial PM$_{2.5}$ estimated model.

*[Please insert Figure 2 here]*

**3.3 Model validation**

In this study, the widely used 10-fold cross-validation (10-CV) procedure (Rodriguez et al., 2010) is
selected for model validation, where all data samples are divided into ten subsets randomly, and nine of
them are used as the training data and the remaining is the testing data, indicating that the training and
testing data are totally independent. This approach is repeated in turn for ten times. Then the error rate
of each test is calculated, and the mean error rate from ten tests determines the final result. Here, the
out-of-sample and out-of-station 10-CV procedures are involved, which the former one is performed
based on the observations and used to evaluate the overall accuracy of the STET model. However, the
later one is performed based on the monitoring stations and used to evaluate the model spatial
performance. This means that training and testing are made of different spatial points, and the





relationship between spatial predictors and PM$_{2.5}$ concentrations estimated in the training dataset is then

predicted on the testing.

## 4. Results and discussion

### 4.1 Validation of MAIAC product

MAIAC AOD retrievals are first evaluated with surface observations using the spatiotemporal matching

approach (Wei et al., 2019b) at 18 AERONET monitoring stations in China (Figure 3). The MAIAC

AOD retrievals show great performance with small estimation errors across mainland China (Figure 2a)

and more than 84% of the matchups satisfy the MODIS expected errors (Levy et al., 2013) at the

national scale. Besides vegetated surfaces, e.g., cropland and grassland, the MAIAC algorithm shows a

considerable accuracy over heterogeneous urban surfaces (Figure 2b). MAIAC AOD products are more

accurate and less biased than the widely used Dark Target (DT) and Deep Blue products at coarse

spatial resolutions (N. Liu et al., 2019; Wei et al., 2018; Tao et al., 2019; Zhang et al., 2019). More

importantly, the DT algorithm cannot be applied with a large amount of missing values over bright

surfaces, and aerosol loadings are significantly overestimated over heterogeneous urban surfaces (Levy

et al., 2013; Wei et al., 2018; 2019c). Therefore, the higher data-quality and spatial-resolution MAIAC

products, which can generate more accurate and detailed PM$_{2.5}$ estimates, are selected in this study.

*[Please insert Figure 3 here]*

### 4.2 Model performance

**4.2.1 Spatial-scale validation**

Figure 4 shows the sample-based and station-based 10-CV results of daily PM$_{2.5}$ estimates for the

traditional ET model and our new developed STET model at the national scale in 2018. The results

suggest that the original ET model works well in estimating PM$_{2.5}$ concentrations with an average out-

of-sample CV-R$^2$, of 0.84 and overall small estimation uncertainties. However, when consider the

spatiotemporal information, the model performance has been significantly improved with an increasing

sample-based CV-R$^2$ equal to 0.89, a stronger regression line (e.g., slope = 0.86), and decreasing RMSE





(~12.46 μg/m$^3$), MAE (~8.26 μg/m$^3$), and MRE (~28.09%) values. Nevertheless, the PM$_{2.5}$

concentrations tend to be overall underestimated at high polluted days (PM$_{2.5}$ > 100 μg/m$^3$) by the

STET model. For the spatial performance, compared to the original ET model, the STET model shows a

stronger spatial predictive power with a higher out-of-station CV-R$^2$ of 0.88, a lower RMSE of 10.97

μg/m$^3$, MAE of 7.17 μg/m$^3$, and MRE of 23.77%. These results illustrate that spatiotemporal

information are crucial in improving the PM$_{2.5}$-AOD relationships and should be carefully considered

when introducing statistical regression models using remote sensing techniques.

*[Please insert Figure 4 here]*


Figure 5 shows the sample-based 10-CV results of the STET model in PM$_{2.5}$ daily estimates over

eastern and western China (according to the widely used Heihe-Tengchong line), and four typical local

regions (Figure 1). The STET model performs differently over eastern and western China mainly due to

significant differences in land cover and climate conditions. There are 1289 uniformly distributed PM$_{2.5}$

stations in eastern China, and 127,241 daily samples were collected. The STET model performs well

eastern China with a high sample-based CV-R$^2$ equal to 0.90 and low estimation uncertainties, i.e.,

RMSE = 9.77 μg/m$^3$, MAE = 6.44 μg/m$^3$, and MRE = 19.24%. By contrast, there are 294 unevenly and

sparsely distributed PM$_{2.5}$ stations in western China, thus about three times fewer daily PM$_{2.5}$ estimates

were collected. The model performance is overall poorer (e.g., CV-R$^2$ = 0.86, and RMSE = 11.99

μg/m$^3$) than over eastern China. This mainly contributed to brighter surfaces (e.g., desert and bare land)

with little vegetation coverage and harsh meteorological conditions over western China.

There were 33,733, 15,199, 6,209, and 6,470 daily samples collected from 233, 184, 95, and 107

uniformly distributed PM$_{2.5}$ monitoring stations in North China Plain (NCP), Yangtze River Delta

(YRD), Pearl River Delta (PRD) and Sichuan Basin (SCB), respectively. For former three typical urban

agglomerations where people closely concerned, the estimated PM$_{2.5}$ concentrations are highly

consistent with surface measurements (CV-R$^2$ = 0.89–0.92) with overall low estimation uncertainties

(i.e., RMSE = 7–12 μg/m$^3$, MAE = 5–8 μg/m$^3$, and MRE = 15–19%). In addition, the STET model also

performs well over Sichuan Basin with an average CV-R$^2$ value equal to 0.87 and comparable



estimation uncertainties to North China Plain. In general, despite some differences in model

performance, the STET model shows an overall good ability in PM$_{2.5}$ estimates at the regional scale.

*[Please insert Figure 5 here]*

The national- and regional-scale aggregated evaluations mainly illustrate the overall performance of the

STET model in PM$_{2.5}$ estimates, however, due to the inhomogeneity of PM$_{2.5}$ monitoring stations, an

additional validation for each monitoring station in China is performed (Figure 6). For statistical

significance, only these monitoring stations with more than ten data samples are plotted. The daily

PM$_{2.5}$ estimations are well related to surface measurements at most individual stations across China.

The average sample-based CV-R$^2$ is 0.84, and the CV-R$^2$ values are higher than 0.8 at more than 73%

of the monitoring stations, especially for eastern China. However, relatively poorer performances (CV-

R$^2$ < 0.6) are observed at some scattered sites located in southwestern and southeastern China. In

general, the STET model shows overall low estimation uncertainties at most sites with average RMSE

and MAE values of 9.3 and 6.5 μg/m$^3$, especially for southern China. Moreover, the average RMSE and

MAE values are < 10 μg/m$^3$ at more than 68% and 93% of the monitoring stations across China. Note

that these stations show larger RMSE values (> 10 μg/m$^3$) in central China mainly due to high polluted

levels. In addition, the average MRE value is 20.88%, and most stations (> 86%) have low MRE values

< 30% in PM$_{2.5}$ estimations in China, especially for those located in eastern and southern China.

*[Please insert Figure 6 here]*

### 4.2.2 Temporal-scale validation

Figure 7 shows the STET model performance from all available monitoring stations in China as a

function of the day of year. The number of data samples in one day ranges from 54 to 1155 with an

average of 466 in 2018. In general, the STET model shows great performance (average CV-R$^2$ = 0.76)

at most days in the year, and more than 76% of the days have CV-R$^2$ values greater than 0.7. Two main

uncertainty metrics, i.e., RMSE and MAE, show similar temporal variations during the year, first

decreasing until around day 250 then gradually increasing. In general, approximately equal 92% of the





days have low RMSE and MAE values less than 15 and 10 μg/m³ over the year. Large estimation uncertainties always occur at the beginning and end of the year mainly due to intense human activities and harsh natural environment. Furthermore, MRE is relatively stable ranging from 13% to 52% with an average value of 23.29%, and more than 87% of the days yield low MRE values less than 30% in

China. These results illustrate that the STET model show great performance in capturing $PM_{2.5}$ concentrations on most days of the year.

*[Please insert Figure 7 here]*

Figure 8 shows sample-based cross-validation results for $PM_{2.5}$ daily estimates divided by four seasons

in 2018 across China. The results suggest that there are obvious differences in model performance at the seasonal level. The STET model performs best in autumn with the highest CV-$R^2$ value of 0.90 and strongest regression line (i.e., slope = 0.88, and intercept = 4.88 μg/m³). The average RMSE, MAE and MRE values are 9.01 μg/m³, 5.87 μg/m³, and 21.10 %, respectively. By contrast, the STET model performs worst in summer with the lowest CV-$R^2$ of 0.76 and smallest slope of 0.74, indicating obvious

underestimations. However, summer shows the least amount of air pollution with most daily $PM_{2.5}$ values < 80 μg/m³, leading to smallest estimation uncertainties. The main reason is that the meteorological conditions in place in summer accelerated the diffusion of pollutants but complicated the $PM_{2.5}$-AOD relationships. The air quality is about two or three times worse in spring and winter than in winter with wider $PM_{2.5}$ ranges and larger standard deviations. Moreover, the STET model shows

similar performances in these two seasonal with almost equal CV-$R^2$ and slope values, as well as close estimation uncertainties.

*[Please insert Figure 8 here]*

### 4.2.3 Predictive power

To test the predictive power of the STET model, the model built for the year of 2018 is used to predict the daily $PM_{2.5}$ concentrations in 2017, then validated against the ground measurements from 2017. This approach can ensure the data samples for model training and validation are completely





independent in both spatial and temporal scales. Figure 9 shows the validation of PM$_{2.5}$ predictions in 2017 at different temporal scales across China. The results show that the STET model can correctly

capture more than 60% of the historical daily PM$_{2.5}$ concentrations (N = 17,7616). The monthly (N = 12,408), seasonal (N = 5,227) and annual (N = 1,461) means of PM$_{2.5}$ predictions are highly correlated with the surface observations with R$^2$ value of 0.79, 0.81, and 0.82, respectively, showing overall small estimation uncertainties (i.e., RMSE < 11.2 μg/m$^3$, MAE < 8.6 μg/m$^3$, MRE < 25.8 μg/m$^3$) across China. These results illustrate that the STET model has a strong predictive power and can well capture

the historical PM$_{2.5}$ concentrations across China.

*[Please insert Figure 9 here]*

### 4.3 Predicted PM$_{2.5}$ maps across China

The monthly PM$_{2.5}$ maps are synthesized and averaged from at least 20% available daily PM$_{2.5}$

estimates for each grid in a month in 2018 across China (Hsu et al., 2012). The monthly PM$_{2.5}$ estimates and ground measurements (N = 12,411) are highly correlated (R$^2$ = 0.94) with a stronger slope of 0.94. The average RMSE and MAE are 5.35 and 3.87 μg/m$^3$, respectively. The monthly spatial coverage varies from 73 to 92%, with an average of 83% across China. The highest (lowest) spatial coverage occcurs around October (January) of the year. Similarly, the monthly mean PM$_{2.5}$ values vary

conversely from 21.2 to 45.1 μg/m$^3$ with the highest (lowest) PM$_{2.5}$ concentration occurring around March (August) of the year.

Figure 10a shows the annual PM$_{2.5}$ maps across China which are generated from monthly PM$_{2.5}$ maps if there are more than eight available values for each grid in 2018 (Wei et al., 2019d). The spatial patterns are similar between the STET-derived 1-km PM$_{2.5}$ map and calculated in-situ measurements (Figure

10b). In addition, validation results suggest that the annual mean PM$_{2.5}$ estimates (N = 1,461) are highly consistent with ground measurements (R = 0.93) with small uncertainties (i.e., RMSE = 3.82 and MAE = 2.90 μg/m$^3$). This illustrate that the synthetic dataset can more accurately reflect the annual PM$_{2.5}$ loadings across China.

The average PM$_{2.5}$ concentration is 33.9±16.3 μg/m$^3$ in 2018 across mainland China. In general, the

most severe PM$_{2.5}$ pollution occurs in the Taklamakan Deseret, where most areas expose high PM$_{2.5}$





concentrations > 80 μg/m$^3$. There are also high-polluted levels over the North China Plain, Sichuan Basin, and Yangtze River Delta, with annual mean PM$_{2.5}$ values of 46.8±11.8, 38.3±10.3, and 37.6±9.4 μg/m$^3$, respectively. These mainly contributed to intensive human activities, special topographic and meteorological conditions. By contrast, the annual mean PM$_{2.5}$ loadings are overall low in the rest areas

of China, e.g., Pearl River Delta (30.5±5.0 μg/m$^3$). However, there may be poor representativeness for these areas over western China with few ground monitoring stations. In general, we have to say that the PM$_{2.5}$ pollution has been significantly reduced in 2018 across China due to the effective emission control measures implemented by the Chinese government (Fang et al., 2019; Ma et al., 2019). However, more than 30% of mainland China still experienced high PM$_{2.5}$ levels exceeding the

recommended air quality level (PM$_{2.5}$ > 35 μg/m$^3$).

*[Please insert Figure 10 here]*

Figure 11 shows seasonal mean PM$_{2.5}$ maps, which are averaged from the available monthly values for each grid, in 2018 across China. Preliminary validation against surface measurements suggest that the

seasonal mean PM$_{2.5}$ estimates are in good accuracy (i.e., R$^2$ = 0.94, RMSE = 4.72 μg/m$^3$, and MAE = 3.49 μg/m$^3$), which can better describe the seasonal variations in PM$_{2.5}$ concentrations across China. There are noticeable spatial differences in PM$_{2.5}$ distributions on the seasonal scale. In winter and spring, more than 77% and 66% of mainland China exposing the high PM$_{2.5}$ levels > 30 μg/m$^3$, yielding poorer air quality. By contrast, PM$_{2.5}$ pollution is slighter in summer and autumn with more than 91%

and 81% of mainland China experiencing low PM$_{2.5}$ levels below the acceptable air quality level. Note that in spring, PM$_{2.5}$ concentrations are particularly high in Xinjiang province due to frequent sand and dust episodes in 2018.

*[Please insert Figure 11 here]*

### 4.4 Comparison with related studies

There is an increasing number of studies on estimating PM$_{2.5}$ using satellite AOD products from local to national scales across China. However, limited by the operational satellite aerosol products, PM$_{2.5}$ can only be estimated at coarse spatial resolutions of approximately 6–10 km (Fang et al., 2016; Li et al.,





2017b; Yu et al., 2017; Chen et al., 2018; Ma et al., 2019; Yao et al., 2019). Recently, with the release
of MODIS 3-km DT aerosol products, the PM$_{2.5}$ estimates can be improved to 3-km spatial resolution
across China (You et al., 2016; Li et al., 2017a; He & Huang, 2018; Chen et al., 2019; Xue et al., 2019).
Therefore, in our study, the spatial resolution of PM$_{2.5}$ estimates has been significantly improved by 3–
10 times to 1 km based on the newly released high-quality MAIAC products across mainland China.
For model performance, our newly developed STET model shows much higher accuracy with higher
CV-R$^2$ values, smaller RMSE and MAE values than the statistical regression models (Table 2), e.g., the
timely structure adaptive model (TSAM, Fang et al., 2016) model, the Gaussian model (Yu et al., 2017),
the Generalized Additive Model (GAM, Chen et al., 2018) model, and the GWR model (Ma et al.,
2014; You et al., 2016), and the GTWR model (He and Huang, 2018). The STET model can also
outperform most machine learning (ML) and deep learning approaches including the RF model (Chen et
al., 2018; Wei et al., 2019e), the XGBoost model (Chen et al., 2019), the Geo- BPNN, GRNN and deep
brief network (DBN) models (Li et al., 2017a, 2017b), and some optical combined models, e.g., the
Daily-GWR (D-GWR) model (He and Huang, 2018), the two-stage model (He and Huang, 2018; Ma et
al., 2019; Yao et al., 2019), and the ML + GAM model (Xue et al., 2019). In addition, there are only a
hanful of studies on the predictive power in PM$_{2.5}$ concentrations across China. The comparison results
show that our STET model is superior to those results reported by previous studies, i.e., the two-stage
model (Ma et al., 2019), the GTWR model (He and Huang, 2018), the ML + GAM model (Xue et al.,
2019), and the STRF model (Wei et al., 2019e).

*[Please insert Table 2 here]*

## 5. Summary and conclusion

With the increase in air pollution over recent years, abundant studies on estimating PM$_{2.5}$ have been
performed using satellite remote sensing. However, most of the PM$_{2.5}$ estimates are reported at spatial
resolutions of 3–10 km, which is inadequate for monitoring air quality at urban areas. The accuracy of
PM$_{2.5}$ estimates is also limited by traditional models. Therefore, we try to generate high-quality PM$_{2.5}$
maps at 1-km higher spatial resolution across China. For this, a new space-time extremely randomized



trees (STET) approach is developed to minimize the spatiotemporal heterogeneities in PM$_{2.5}$ and

improve the estimate accuracy.

Our results suggest that the STET model shows great performance in estimating daily PM$_{2.5}$

concentrations with a relatively high sample-based cross-validation coefficient of 0.89, low RMSE of

10.35 μg/m$^3$, MAE of 6.71 μg/m$^3$ and MRE of 21.37% at the national scale. Comparisons illustrate that

spatiotemporal information is of great importance and should be carefully considered during model

development. The STET model shows better performance at most monitoring stations and individual

days in the year. The North China Plain and the Sichuan Basin regions, under the influence of intense

human activities and poor dispersion conditions, have high PM$_{2.5}$ loadings. Moreover, the STET model

can outperform most models presented in previous related studies in terms of spatial resolution, model

accuracy and predictive power. This study suggests that the 1-km-resolution PM$_{2.5}$ dataset will be of

great importance in future atmospheric pollution focused on medium- or small-scale areas. In addition,

the STET model will be applied to produce the historical PM$_{2.5}$ dataset across China in our future

studies since MODIS can cover global observations nearly over the past 20 years.

**Data availability**

Data are available by contacting the author (weijing_rs@163.com).

**Author contributions**

ZL designed the research, and JW carried out the research and wrote the initial draft of this manuscript.

WX, LX, LL, HW, and YS helped collected and processed the used data. ZL, LS, JG, YP, and JL

helped review the manuscript. All authors made substantial contributions to this work.

**Competing interests**

The authors declare that they have no conflict of interest.




**Acknowledgements**

The in-situ $PM_{2.5}$ measurements are available from the China National Environmental Monitoring Center (http://www.cnemc.cn). The MODIS series products are available at https://search.earthdata.nasa.gov/, and the ERA-Interim reanalysis products are available at

https://www.ecmwf.int/en/forecasts/datasets/reanalysis-datasets/era-interim. The AERONET measurements are available at https://aeronet.gsfc.nasa.gov/.

**Financial support**

This research has been supported by the National Key R&D Program of China (2017YFC1501702), the

National Natural Science Foundation of China (91544217), the U.S. National Science Foundation (AGS1534670), and the BNU Interdisciplinary Research Foundation for the First-Year Doctoral Candidates (BNUXKJC1808).

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



595                  Table 1. Summary of the data sources used in this study.

| Dataset | Variable | Content | Unit | Spatial Resolution | Temporal Resolution | Data source |
|---|---|---|---|---|---|---|
| PM$_1$ | PM$_{2.5}$ | PM$_{2.5}$ | µg/m$^3$ | - | Hourly | CNEMC |
| AOD | AOD | MAIAC AOD | - | 1 km ×1 km | Daily | MCD19A2 |
| Meteorological data | BLH | Boundary layer height | m | 0.125°×0.125° | 3-hour | ERA-Interim reanalysis product |
| | PRE | Total precipitation | mm | 0.125°×0.125° | 3-hour | |
| | EP | Evaporation | mm | 0.125°×0.125° | 3-hour | |
| | RH | Relative humidity | % | 0.125°×0.125° | 3-hour | |
| | TEM | 2-m air temperature | K | 0.125°×0.125° | 6-hour | |
| | SP | Surface pressure | hPa | 0.125°×0.125° | 6-hour | |
| | WS | 10-m wind speed | m/s | 0.125°×0.125° | 6-hour | |
| | WD | 10-m wind direction | m/s | 0.125°×0.125° | 6-hour | |
| Land cover | NDVI | NDVI | - | 500 m × 500 m | Monthly | MOD13A3 |
| | LUC | Land use cover | - | 500 m × 500 m | Annually | MCD12Q1 |
| Topography | DEM | DEM | m | 90 m × 90 m | - | SRTM |
| | Relief | Surface relief | m | 90 m × 90 m | - | |
| | Aspect | Surface aspect | degree | 90 m × 90 m | - | |
| | Slope | Surface slope | degree | 90 m × 90 m | - | |
| Population | NTL | Night lights | W/cm$^2$/sr | 500 m × 500 m | Monthly | VIIRS |





Table 2. Comparison between model performances of the STET model and other models from previous related studies focused on China.

| Model | Resolution | Model Validation | | | Predictive power | | Literature |
|---|---|---|---|---|---|---|---|
| | | $R^2$ | RMSE | MAE | Daily | Monthly | |
| GWR | 10 km | 0.64 | 32.98 | 21.25 | - | - | Ma et al., (2014) |
| TSAM | 10 km | 0.80 | 22.75 | 15.99 | - | - | Fang et al. (2016) |
| Gaussian | 10 km | 0.81 | 21.87 | - | - | - | Yu et al. (2017) |
| RF | 10 km | 0.83 | 18.08 | - | - | - | Chen et al. (2018) |
| GAM | | 0.55 | 29.13 | - | - | - | |
| Geo-BPNN | 10 km | 0.84 | 15.23 | 10.34 | - | - | Li et al. (2017b) |
| Geo-GRNN | | 0.82 | 16.93 | 12.34 | - | - | |
| Geo-DBN | | 0.88 | 13.03 | 08.54 | - | - | |
| Two-stage | 10 km | 0.77 | 17.10 | 11.51 | 0.41 | 0.73 | Ma et al. (2019) |
| Two-stage | 6 km | 0.60 | 21.76 | 14.41 | - | - | Yao et al. (2019) |
| GRNN | 3 km | 0.67 | 20.93 | 13.90 | - | - | Li et al. (2017a) |
| GWR | 3 km | 0.81 | 21.87 | - | - | - | You et al. (2017) |
| D-GWR | 3 km | 0.72 | 21.01 | 14.59 | - | - | He & Huang (2018) |
| Two-stage | | 0.71 | 21.21 | 13.50 | - | - | |
| GTWR | | 0.80 | 18.00 | 12.03 | 0.41 | | |
| XGBoost | 3 km | 0.86 | 14.98 | - | - | - | Chen et al. (2019) |
| ML + GAM | 3 km | 0.61 | 27.80 | 17.70 | 0.57 | 0.74 | Xue et al. (2019) |
| STRF | 1 km | 0.85 | 15.57 | 9.77 | 0.55 | 0.73 | Wei et al. (2019e) |
| STET | 1 km | 0.89 | 10.35 | 6.71 | 0.60 | 0.80 | Our study |




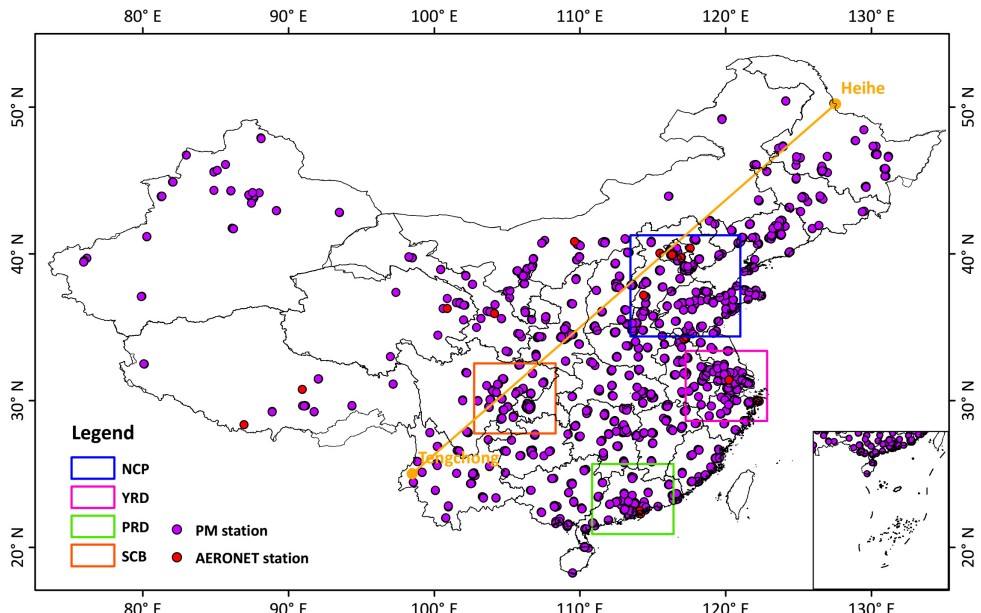

Figure 1. Spatial distributions of PM$_{2.5}$ and AERONET monitoring stations in China. The Heihe-Tengchong line (orange line) shows the boundary between Eastern and Western China.





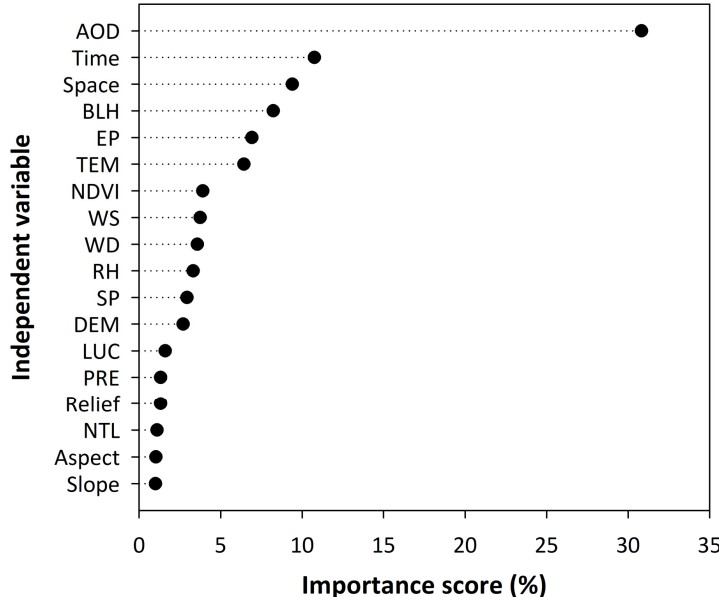


Figure 2. Importance score (%) of independent variables to PM$_{2.5}$ estimates for the STET model.





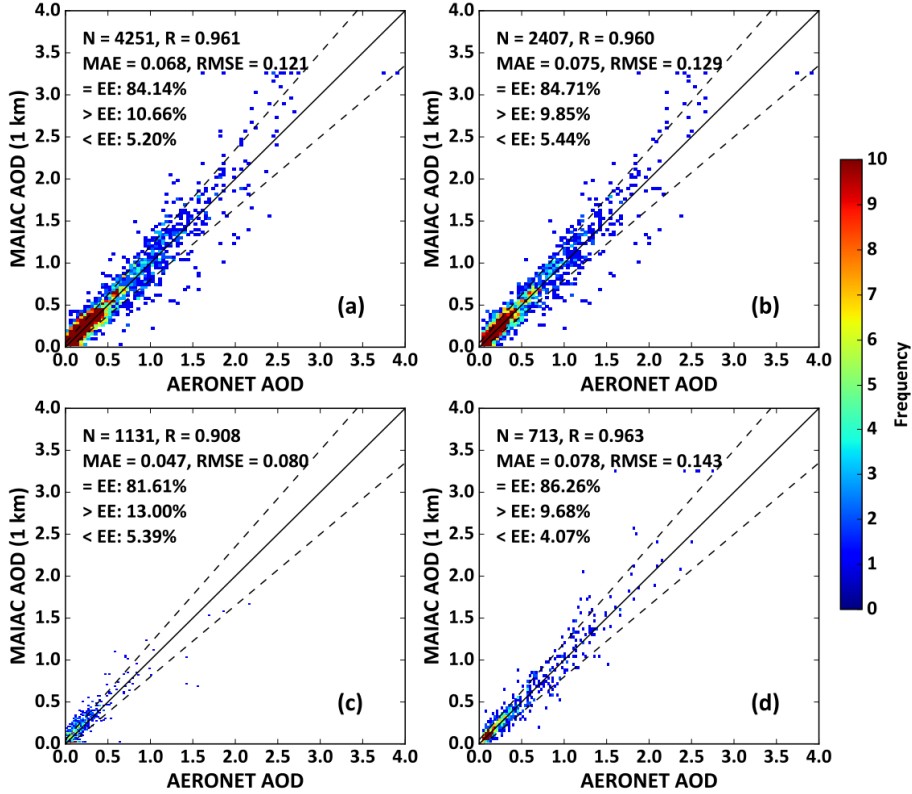

Figure 3. Scatter plots of MAIAC AOD retrievals versus AERONET AODs at 550 nm in (a) China, and (b) urban, (c) cropland, and (d) grassland. The dotted lines represent the upper and lower boundaries of the expected error (EE).




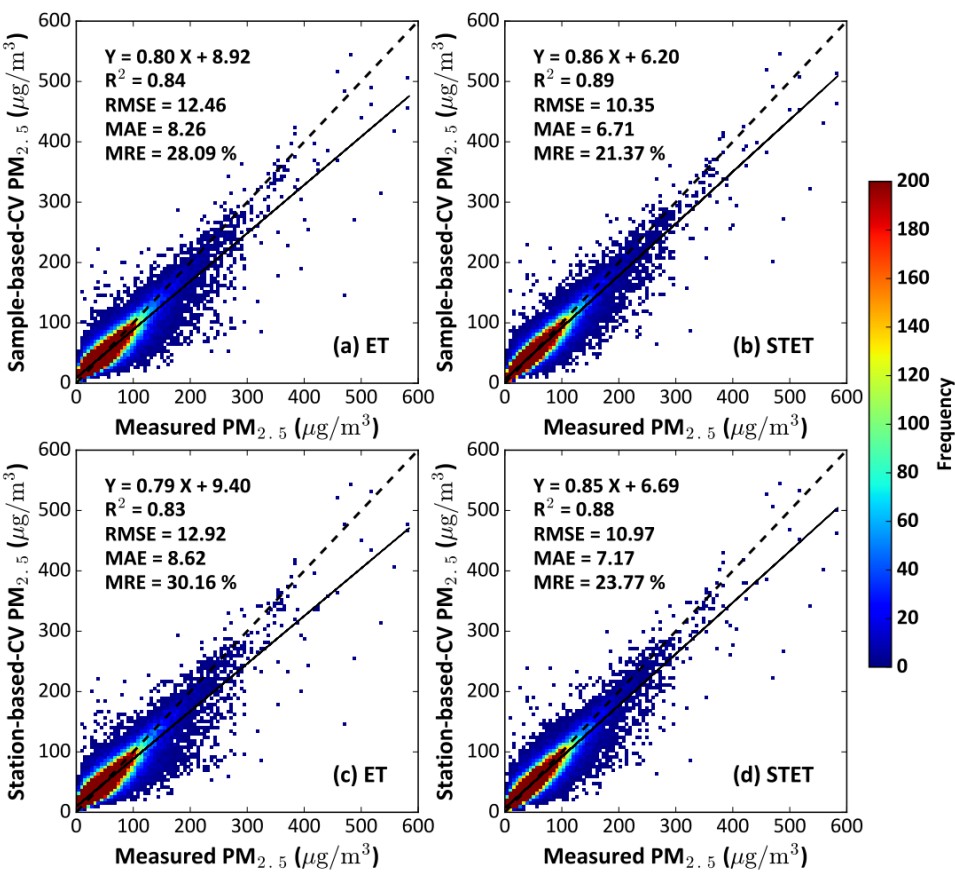

Figure 4. Density scatter plots of sample-based (top row) and station-based (bottom row) 10-CV results for the ET and STET models at the daily level (N = 167,692) in 2018 across mainland China.





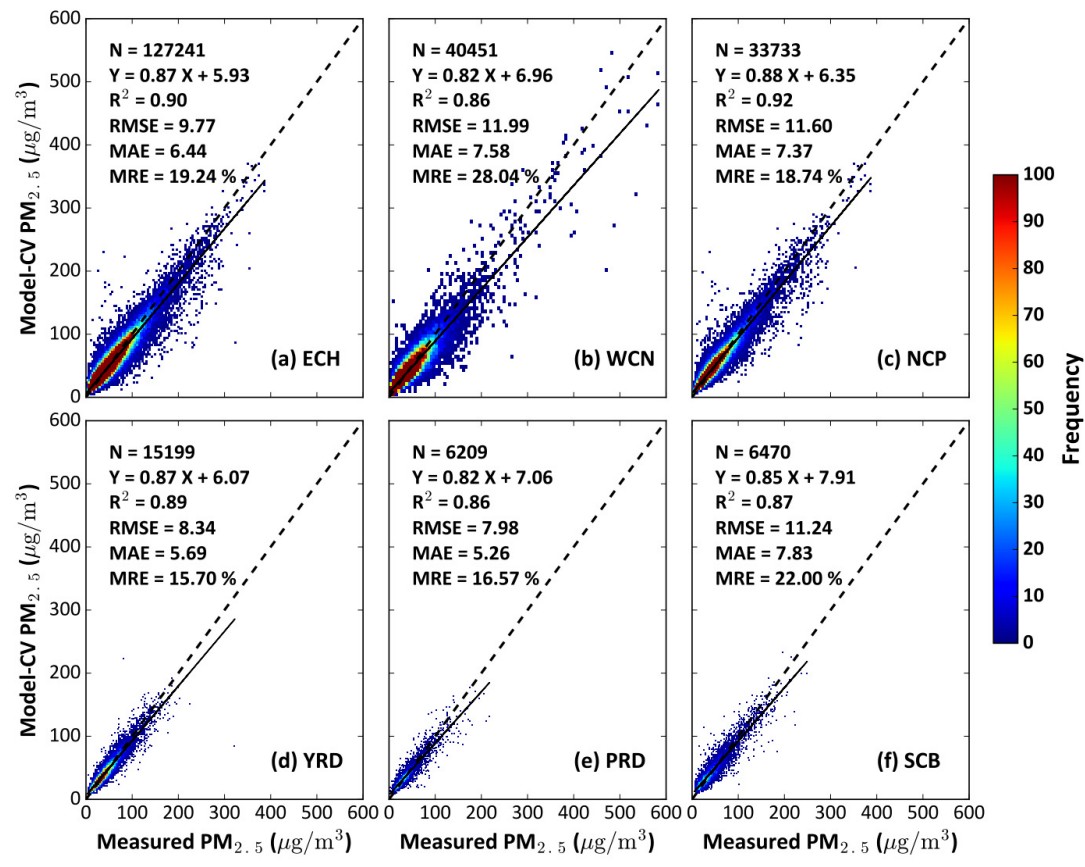

Figure 5. Density scatter plots of sample-based 10-CV results for (a) eastern China (ECH), (b) western China (WCH), (c) North China Plain (NCP), (d) Yangtze River Delta (YRD), (e) Pearl River Delta (PRD), and (f) Sichuan Basin (SCB) in 2018.



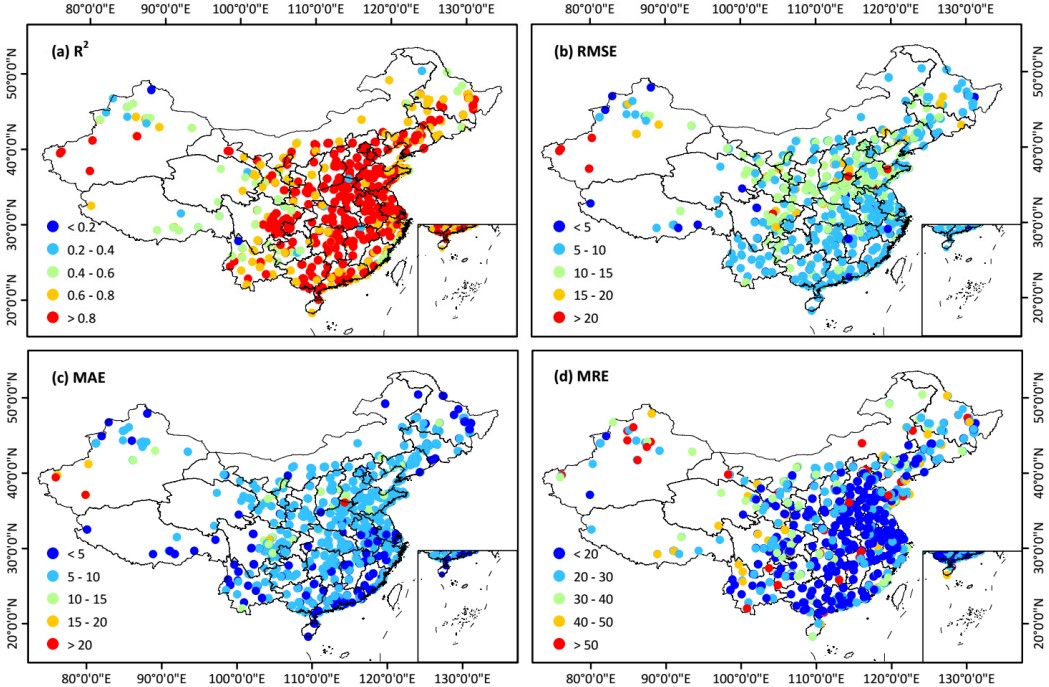

Figure 6. Spatial distributions of the site-scale performance of the STET model for (a) the sample-based CV-$R^2$, (b) RMSE, (c) MAE, and (d) MRE in 2018 across China.



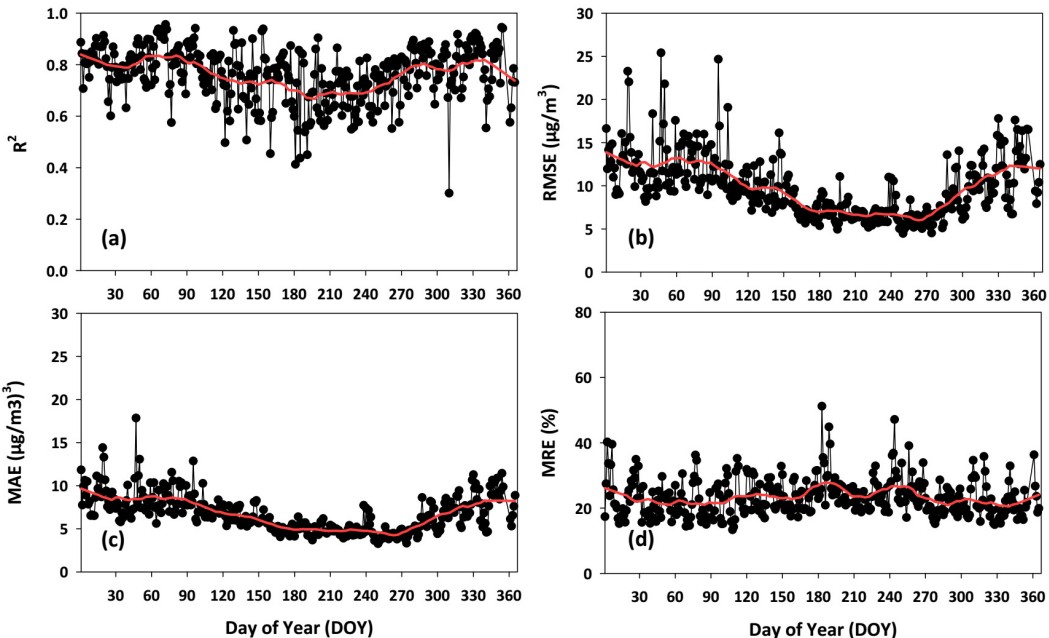

Figure 7. Time series of the daily performance of the STET model in terms of (a) sample-based CV-R$^2$,
(b) RMSE, (c) MAE, and (d) MRE in 2018 across China





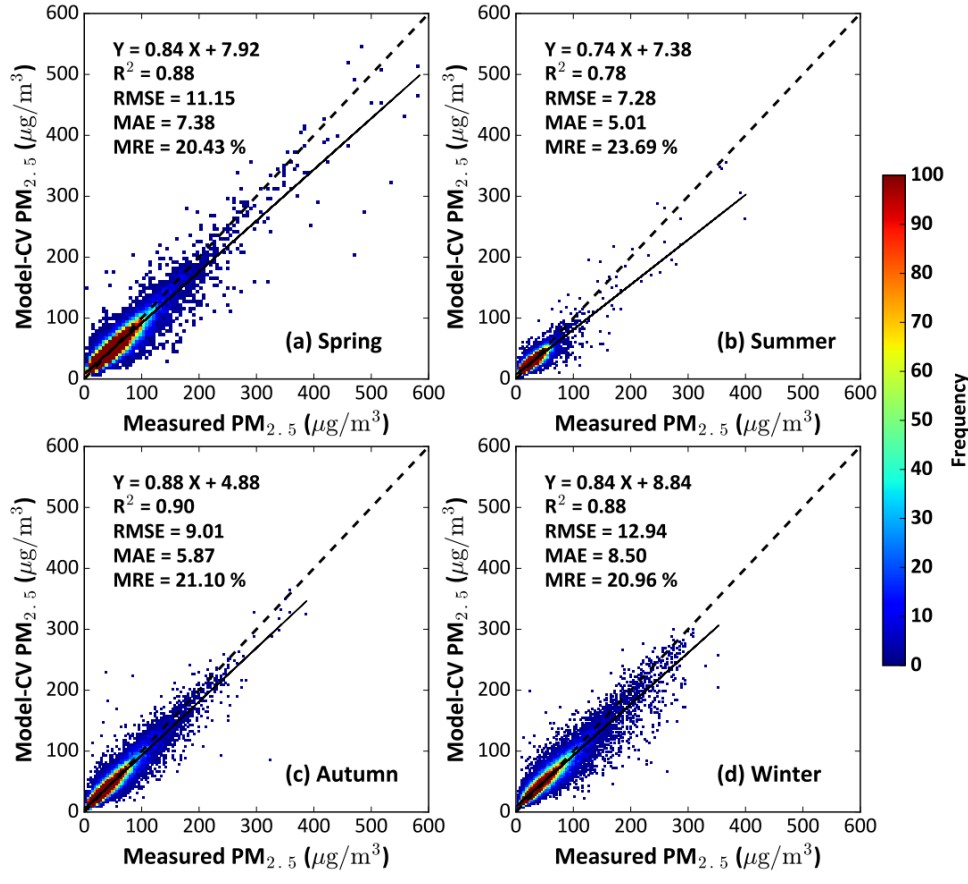

Figure 8. Density scatter plots of sample-based 10-CV results for the STET model for four seasons in 2018 across China.





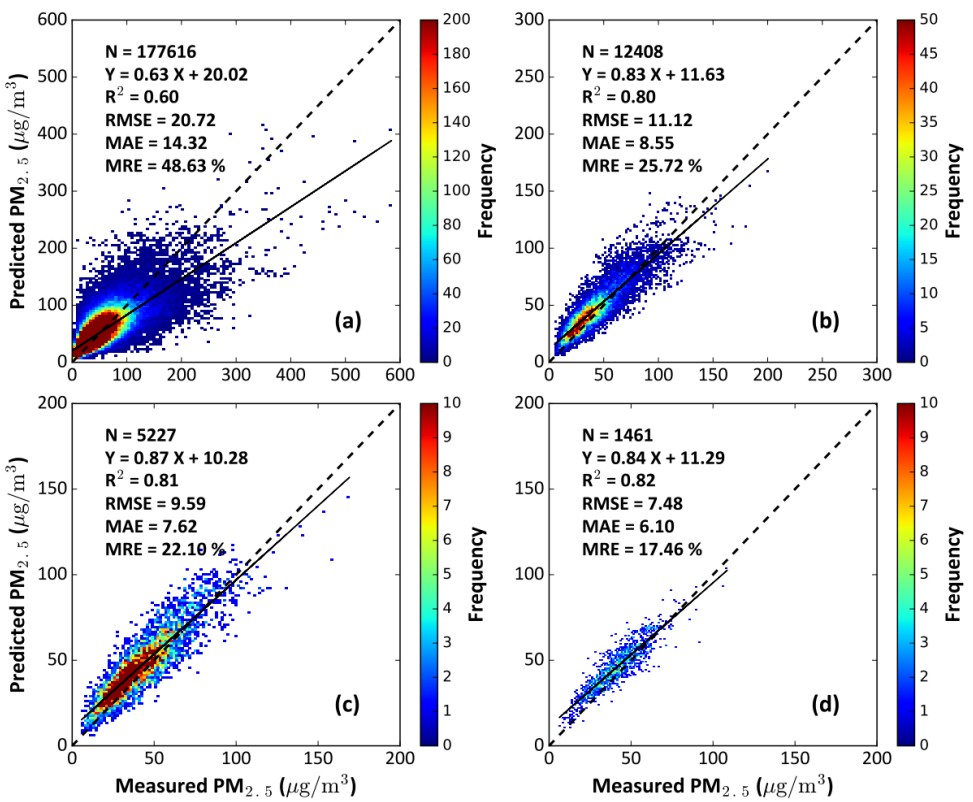

Figure 9. Density scatter plots of sample-based 10-CV results for the STET model for four seasons in
630                                                    2018 across China.





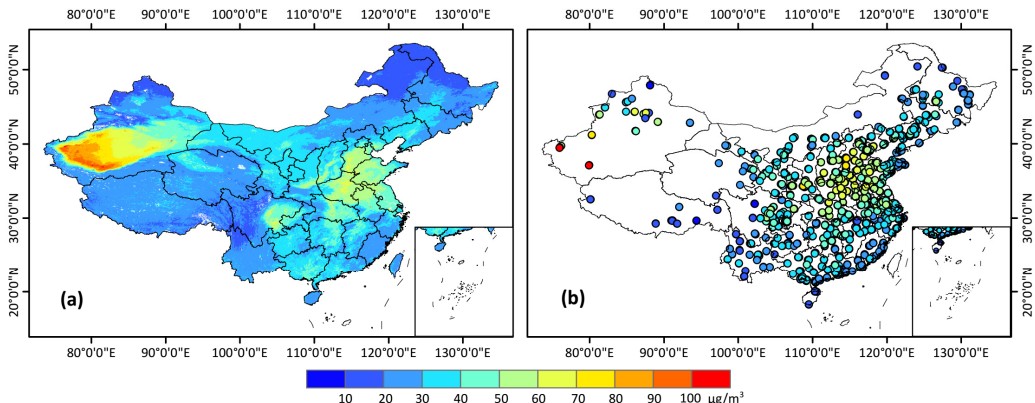

Figure 10. Spatial distributions of annual mean (a) PM$_{2.5}$ estimates and (b) surface observations in 2018 across China.





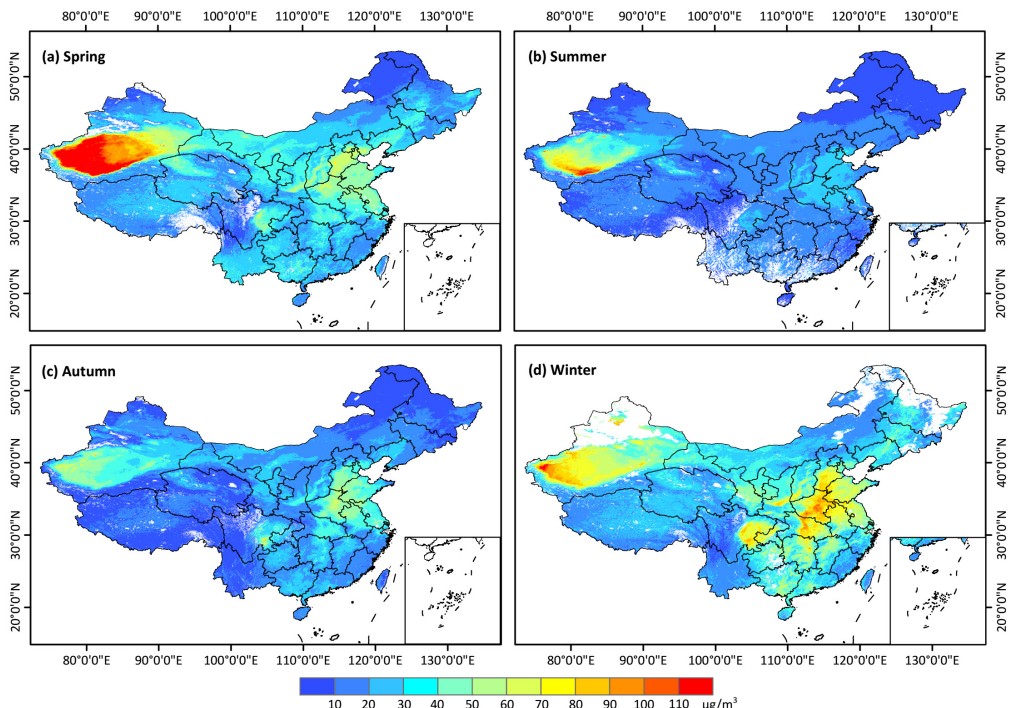

Figure 11. Spatial distributions of seasonal mean 1-km-resolution PM$_{2.5}$ concentrations for four seasons
in 2018 across China.