# Peer review of "Improved 1-km-resolution PM2.5 estimates across China using enhanced space-time extremely randomized trees"

_Atmospheric Chemistry and Physics, 2019_

## Referee Comment (RC1) · Anonymous Referee #2 · 2 Nov 2019

This study built a new space-time extremely randomized trees model (STET), which integrates information from satellite-based aerosol optical depth (AOD) measurements, ground-based PM2.5 observations, and other auxiliary data (e.g., meteorological data), to retrieve daily surface PM2.5 concentrations over China. The newly-developed model outperforms most of the previously reported models in capturing the spatiotemporal variations in surface PM2.5 concentrations and in finer spatial resolution. Overall, this manuscript is well organized with extensive evaluations on the model performance. There are some minor concerns that should be addressed before publication.

1. Eq. 1. It is not clear to me how the authors apply these equations. Did the authors

apply the relationships between Terra- and Aqua-based AOD measurements to fill the missing AOD value for one sensor while another sensor has a valid measurement on the same day? Please clarify the usage of Eq. 1.

2. L201-202. It is possible that the limited impact of precipitation on PM2.5 estimates can be attributed to the fact that there's a high probability of missing AOD measurements on rainy days?

3. It is unclear to me how the authors compare monthly, seasonal, and annual mean PM2.5 retrievals with observed PM2.5 data. For example, for one grid with 100 days of valid daily PM2.5 retrieval, to compare annual mean PM2.5 retrieval with observation, did the authors calculate the corresponding 100-day mean PM2.5 observation or the 365-day mean PM2.5 observation for comparison?

4. L247-248. What's the reason for the overall underestimation of PM2.5 concentration in high polluted days by the STET model?

5. L310-316. What's the possible impact of variations in the valid sample number of AOD measurement across seasons on the differences in model performance at the seasonal level?

6. L361-363. Results in this study cannot support the conclusion here (i.e., air quality improvement from clean air policies) as only one-year PM2.5 concentration data was developed. Please rephrase this sentence.

7. The caption for Fig.9 is incorrect.

8. L36. "cross-validation coefficient" is unclear here, please clarify whether it means correlation coefficient (R) or coefficient of determination (R2).

8. Would suggest spelling out all statistical metrics (e.g., R2, RMSE, MAE, MRE) when you first mention them.

9. Would suggest thoroughly checking the manuscript to avoid grammar errors and

make the manuscript more readable.

---

## Referee Comment (RC2) · Anonymous Referee #1 · 6 Nov 2019

Using the newly-developed space-time extremely randomized trees (STET) model, this study is aimed at estimating the 1-km-resolution $PM_{2.5}$ surface concentrations across China. Besides meteorology, land surface conditions and population, a space term and a time term representing the spatial autocorrelation and temporal variation of $PM_{2.5}$, respectively are also included to derive the $PM_{2.5}$-AOD relationship. Overall this manuscript is well written, and potentially improves our understanding regarding how to retrieve the $PM_{2.5}$ concentrations from AOD products and other auxiliary data. However, before I recommend this manuscript to be published, the authors should carefully address and clarify my several comments.

General comments:

1.  The relationship between (surface layer) $PM_{2.5}$ and AOD might largely depend on the compositions (including aerosol water, as Reddington et al. (2019) indicated that aerosol water uptake and hygroscopic growth would also impact the AOD), vertical profile and size distribution of $PM_{2.5}$. Thus I find that some results in Figure 2 are confusing, and needs further analysis and clarification: 1) In Section 3.2, it is unclear that how the importance scores of all selected independent variables and spatiotemporal information to $PM_{2.5}$ estimates for the STET model are calculated. 2) Why RH turns out to be a much less important parameter, and it has an importance score that is only slightly higher than those negligible parameters do. RH is an important factor determining the aerosol compositions and water uptake, and recent air quality studies (e.g., Sun et al., 2014;Zheng et al., 2015) showed that high RH conditions facilitate rapid production of secondary PM. 3) Furthermore, the parameter of precipitation could significantly impact the removal of PM, but is negligible in the STET model. Both RH and precipitation are associated with cloud, and what is the uncertainty for the predicted $PM_{2.5}$-AOD relationship caused by the treatment of AOD data on cloudy dates?

2.  The authors declared that STET model exhibited a strong predictive power and could be used to predict the historical $PM_{2.5}$ records in the Abstract Section (in Line 39). This conclusion could be inappropriate as the authors only tested the year of 2017. Emissions were not expected to change greatly between 2017 and 2018. Actually I doubt the applicability for the STET model. The space and time terms seem confusing to me, and the former term is represented by the geographical difference between two pixels, while the latter term is represented by the difference for a given pixel on different days in a year. I think they might be "residual terms" to implicitly resolve the "unknown parts" unexplained by other independent parameters. I mean, the authors need more independent parameters that could explicitly explain the $PM_{2.5}$ compositions, vertical profile and size distribution. Why not emissions for different precursors (e.g., $SO_2$, $NO_x$ and VOCs) as well as fine size dust are included as independent parameters?

3.  Equation 1 is confusing. The authors mean:

$$\tau_T = k_1 \tau_{A,original} + b_1 \quad \text{and} \quad \tau_A = k_2 \tau_{T,original} + b_2 ?$$

What is the $R^2$ for each linear regression? Are these two linear regressions consistent with each other? Why not to average the Terra and Aqua data directly?

4.  The description for the STET method in Section 3 is not readily to understand. Please add clarification (better to include a schematic) so that ACP readers with less experiences in machine learning could generally understand the fundamentals of the STET method.

5.  In Figure 7, what is surprising is that I see a good positive correlation pattern between R and RMSE. Generally a good model performance is associated with a high R and a low RMSE against

observations. Please check and clarify.

Specific comments:

1. Line 48, the "evenly dispersed" is confusing, and is conflict with the "PM$_{2.5}$ shows great spatial and temporal heterogeneities" in Line 80.
2. Line 175, better replace "differences" by variation.
3. Line 227, typos: Figure 2 or Figure 3?
4. Line 247, what is definition for MAE and MRE?
5. Figure 9, typos: the year is 2018 or 2017? Also please add the season labels for each plot.

Reddington, C. L., Morgan, W. T., Darbyshire, E., Brito, J., Coe, H., Artaxo, P., Scott, C. E., Marsham, J., and Spracklen, D. V.: Biomass burning aerosol over the Amazon: analysis of aircraft, surface and satellite observations using a global aerosol model, Atmos. Chem. Phys., 19, 9125-9152, 10.5194/acp-19-9125-2019, 2019.

Sun, Y., Jiang, Q., Wang, Z., Fu, P., Li, J., Yang, T., and Yin, Y.: Investigation of the sources and evolution processes of severe haze pollution in Beijing in January 2013, Journal of Geophysical Research: Atmospheres, 119, 4380-4398, 2014.

Zheng, G., Duan, F., Su, H., Ma, Y., Cheng, Y., Zheng, B., Zhang, Q., Huang, T., Kimoto, T., and Chang, D.: Exploring the severe winter haze in Beijing: the impact of synoptic weather, regional transport and heterogeneous reactions, Atmos. Chem. Phys., 15, 2969-2983, 2015.

---

## Referee Comment (RC3) · Anonymous Referee #3 · 8 Dec 2019

I noticed that the same authors published a very similar paper in ES&T, https://pubs.acs.org/doi/10.1021/acs.est.9b03258. The only difference is between PM2.5 and PM1.0. However, the ACP paper needs originality. Moreover, the manuscript has some fatal defects, (1) It does not work well with high pollution events, which is paid more attention. (2) Such method seems falling into a dead cycle, the results were compared by the observations which were used to fit the parameters. I do not think it works with another independent database, Some similar comments were pointed by the othe two reviewers.
* * *
[Figure]

2019.

---

## Author Response (AR1)

**Reviewer: 1**

This study built a new space-time extremely randomized trees model (STET), which integrates information from satellite-based aerosol optical depth (AOD) measurements, ground-based PM2.5 observations, and other auxiliary data (e.g., meteorological data), to retrieve daily surface PM2.5 concentrations over China. The newly-developed model outperforms most of the previously reported models in capturing the spatiotemporal variations in surface PM2.5 concentrations and in finer spatial resolution. Overall, this manuscript is well organized with extensive evaluations on the model performance.

Response: We appreciate the time and effort you spent on this manuscript, and we have carefully revised our manuscript. The responses to the questions raised in your report are as follows.

There are some minor concerns that should be addressed before publication. 1. Eq. 1. It is not clear to me how the authors apply these equations. Did the authors apply the relationships between Terra- and Aqua-based AOD measurements to fill the missing AOD value for one sensor while another sensor has a valid measurement on the same day? Please clarify the usage of Eq. 1.

**Response:** We have replaced the regression method with the average approach according to Reviewer#2's suggestion, and we have clarified this in Section 3.1 of the revised manuscript as follows:

"Terra and Aqua MAIAC AOD retrievals are thus averaged for each pixel on each day to form a new dataset and enlarge the spatial coverage."

2. L201-202. It is possible that the limited impact of precipitation on PM2.5 estimates can be attributed to the fact that there's a high probability of missing AOD measurements on rainy days?

**Response:** Yes, that's the reason for the limited impact of precipitation on PM2.5 estimates. We have added this as "This can be attributed to the high probability of missing AOD retrievals on rainy days." in Section 3.3 of the revised manuscript.

3. It is unclear to me how the authors compare monthly, seasonal, and annual mean PM2.5 retrievals with observed PM2.5 data. For example, for one grid with 100 days of valid daily PM2.5 retrieval, to compare annual mean PM2.5 retrieval with observation, did the authors calculate the corresponding 100-day mean PM2.5 observation or the 365-day mean PM2.5 observation for comparison? **Response:** We compared the monthly, seasonal, and annual mean PM2.5 retrievals with PM2.5 observations using the same number of valid days. We have clarified this in the revised manuscript as follows:

"Synthetized PM2.5 retrievals are validated against PM2.5 surface observations by calculating the effective values from the same number of valid days at monthly, seasonal, and annual time scales (Figure 10)."

4. L247-248. What's the reason for the overall underestimation of PM2.5 concentration in high polluted days by the STET model?

**Response:** We have discussed potential reasons in Section 5.1 in the revised manuscript as follows:

"Potential causes are: 1) There are large estimation errors in AOD retrievals under severe pollution conditions in China (Wei et al., 2019c). This is further rooted to the fundamental limitations of satellite-based AOD retrievals, i.e., the non-linear to reflectance and the high sensitivity of the single-scattering albedo (Z. Li et al., 2009); 2) High AOD does not correspond to high PM2.5 concentrations because their ratio is highly variable over space and time, affected by both natural and human factors; 3) The number of samples for high-pollution cases is small, hindering the ability to train the model."

5. L310-316. What's the possible impact of variations in the valid sample number of AOD measurement across seasons on the differences in model performance at the seasonal level?

**Response:** We have discussed the potential causes for the differences in the number of data samples and model performance at the seasonal level in Section 4.2.2 of the revised manuscript as follows:

"Results suggest that there are clear differences in the number of valid data samples because of the long-term snow/ice cover in winter and more frequent clouds in summer, resulting in an overall smaller number of samples than in the other two seasons. ... The differences in model performance among the seasons are mainly attributed to seasonal variations in natural conditions and human activities. Meteorological conditions in summer favor the diffusion of pollutants but complicate the PM2.5-AOD relationship (Su et al., 2018, 2020), whereas direct emissions of pollutants are greater in winter, resulting in severe air pollution."

6. L361-363. Results in this study cannot support the conclusion here (i.e., air quality improvement from clean air policies) as only one-year PM2.5 concentration data was developed. Please rephrase this sentence.

Response: We have removed this sentence from the manuscript.

7. The caption for Fig.9 is incorrect. **Response:** We have corrected the caption in the revised manuscript.

8. L36. "cross-validation coefficient" is unclear here, please clarify whether it means correlation coefficient (R) or coefficient of determination (R2). **Response:** We have clarified this in the revised manuscript.

9. Would suggest spelling out all statistical metrics (e.g., R2, RMSE, MAE, MRE) when you first mention them. **Response:** Done. 10. Would suggest thoroughly checking the manuscript to avoid grammar errors and make the manuscript more readable.

**Response:** The manuscript has been more carefully edited by a native speaker.

**Reviewer: 2**

Using the newly-developed space-time extremely randomized trees (STET) model, this study is aimed at estimating the 1-km-resolution PM2.5 surface concentrations across China. Besides meteorology, land surface conditions and population, a space term and a time term representing the spatial autocorrelation and temporal variation of PM2.5, respectively are also included to derive the PM2.5-AOD relationship. Overall this manuscript is well written, and potentially improves our understanding regarding how to retrieve the PM2.5 concentrations from AOD products and other auxiliary data. However, before I recommend this manuscript to be published, the authors should carefully address and clarify my several comments.

**Response:** We appreciate the time and effort the reviewer spent on this manuscript and the insightful comments and constructive suggestions. In light of your opinion, we have carefully revised our manuscript. The responses to the questions raised in your report are as follows.

**General comments:**

1. The relationship between (surface layer) PM2.5 and AOD might largely depend on the compositions (including aerosol water, as Reddington et al. (2019) indicated that aerosol water uptake and hygroscopic growth would also impact the AOD), vertical profile and size distribution of PM2.5. Thus, I find that some results in Figure 2 are confusing, and needs further analysis and clarification: 1) In Section 3.2, it is unclear that how the importance scores of all selected independent variables and spatiotemporal information to PM2.5 estimates for the STET model are calculated. **Response:** We agree with you and we have mentioned this in the manuscript and cited the references. In addition, the importance score is described in more detail in the revised manuscript. The importance score of each independent variable used to estimate PM2.5 is calculated based on the Gini index (GI). We have added a more detailed description in Section 3.3 of the revised manuscript as follows:

"... the GI index is selected to calculate the importance score of each independent variable on PM2.5 estimates because of its higher accuracy and stability as a variable importance measure, especially for continuous variables with low signal-to-noise ratios (Jiang et al., 2009; Calle and Urrea, 2011), expressed as

$$GI(\omega) = \sum_{n=1}^{N} \omega_n (1 - \omega_n) = 1 - \sum_{n=1}^{N} \omega_n^2 , \qquad (2)$$

where *n* represents the number of the categories (N = 1, ..., *n*), and  $\omega_n$  represents the sample weight of each category. The importance of one feature ( $X_j$ ) on node *m* is that the GI changes before and after node *m* branching:

$$\Delta GI_{jm} = GI_m - GI_l - GI_r , (3)$$

where  $GI_l$  and  $GI_r$  represent the GI of two new nodes after branching. The importance score for one feature (*ISj*) in then the extra-trees with *k* trees (*i* = 1, ..., *k*), calculated as

$$IS_j = \sum_{i=1}^k \Delta GI_{ij} = \sum_{i=1}^k \sum_{m \in M} \Delta GI_{jm} , (4)$$

where  $\Delta GI_{ij}$  represents the importance of  $X_i$  in the *i*th tree when the node of feature  $X_i$  in decision tree *j* belongs to set *M*. Finally, an additional normalization approach is performed to all obtained importance scores for each feature."

2) Why RH turns out to be a much less important parameter, and it has an importance score that is only slightly higher than those negligible parameters do. RH is an important factor determining the aerosol compositions and water uptake, and recent air quality studies (e.g., Sun et al., 2014; Zheng et al., 2015) showed that high RH conditions facilitate rapid production of secondary PM.

**Response:** We agree with you that RH should have a large influence on the production of PM2.5. However, a potential reason why RH turns out to be less important is that high RH conditions are potentially highly related to cloudy/rainy days, especially in summer, when there is a high probability of missing AOD retrievals. In addition, this importance score only represents the importance of features in splitting during the extra-tree construction, not the contribution of features to PM2.5 in physical mechanisms. We have clarified these in in Section 3.3 of the revised manuscript as follows:

"The PM2.5-AOD relationship might largely depend on the compositions (e.g., aerosol water, Reddington et al., 2019; Jin et al., 2020). High RH conditions and precipitation should have large influences on the production and removal of PM2.5 (Sun et al., 2014; Zheng et al., 2015). However, RH and PRE turn to be less important with overall low importance scores in the STET model, which may be attributed to the fact that aerosol retrieval algorithms only work under cloud-free conditions when RH is relatively low. More importantly, the calculated importance score only represents the importance of features in splitting during the extra-tree construction, not the contribution of features to PM2.5 in physical mechanisms."

3) Furthermore, the parameter of precipitation could significantly impact the removal of PM, but is negligible in the STET model. Both RH and precipitation are associated with cloud, and what is the uncertainty for the predicted PM2.5-AOD relationship caused by the treatment of AOD data on cloudy dates?

**Response:** We agree with you that the precipitation should have a large influence on the removal of PM2.5. However, it shows the lowest important score and is negligible because remote sensing aerosol retrieval algorithms cannot work when clouds are present, so there are no AOD retrievals on rainy days. Similarly, the importance score only refers to the importance of features in splitting during the extra-tree construction and not the contribution of features to PM2.5 in physical mechanisms. We have added this description to Section 3.3 of the revised manuscript (See above comment):

2. The authors declared that STET model exhibited a strong predictive power and could be used to predict the historical PM2.5 records in the Abstract Section (in Line

39). This conclusion could be inappropriate as the authors only tested the year of 2017. Emissions were not expected to change greatly between 2017 and 2018. Actually, I doubt the applicability for the STET model. The space and time terms seem confusing to me, and the former term is represented by the geographical difference between two pixels, while the latter term is represented by the difference for a given pixel on different days in a year. I think they might be "residual terms" to implicitly resolve the "unknown parts" unexplained by other independent parameters. I mean, the authors need more independent parameters that could explicitly explain the PM2.5 compositions, vertical profile and size distribution. Why not emissions for different precursors (e.g., SO2, NOx and VOCs) as well as fine size dust are included as independent parameters?

**Response:** PM2.5 changes dramatically in space, and varies over time, showing significant spatiotemporal heterogeneities and patterns. Thus, introducing the spatial and temporal terms account for the spatiotemporal autocorrelations of PM2.5 between different points for each day and between consecutive time series at the same place. In addition, per your suggestion, we have included emissions for main precursors and fine-sized dust as independent parameters to enhance the STET model and improve the estimation of PM2.5 in Section 3 of the revised manuscript as follows: "Different with our previous study (Wei et al., 2019b), pollutant emissions for different precursors (including SO2, NOx, CO, and volatile organic compounds) and fine-sized dust are also employed to help explicitly explain the PM2.5 composition, collected from a multi-resolution emission inventory for China (Zhang et al., 2007)."

In addition, we have updated and re-described in detail all the results in Sections 3 and 4. Results show that the model performance is overall improved.

3. Equation 1 is confusing. What is the R2 for each linear regression? Are these two linear regressions consistent with each other? Why not to average the Terra and Aqua data directly?

**Response:** We have replaced the regression method with the average approach per your suggestion and clarified this in the revised manuscript as follows:

"Terra and Aqua MAIAC AOD retrievals are thus averaged for each pixel on each day to form a new dataset and enlarge the spatial coverage."

4. The description for the STET method in Section 3 is not readily to understand. Please add clarification (better to include a schematic) so that ACP readers with less experiences in machine learning could generally understand the fundamentals of the STET method.

**Response:** We have added clarification and a schematic of the STET model in Section 3.4 of the revised manuscript as follows:

"For the enhanced STET model, all the selected independent variables are first input into the ERT model, and the random splits (S,  $a_i$ ) are established according to the whole of training data samples; then totally different *K* attributes are selected randomly from all attributes according to spatial and temporal differences; then *K*  random splits are generated  $(s_1, ..., s_k)$ , and a split  $(s^*)$  is selected by calculating the score measure function, i.e., Score $(s^*, S)$ ; then split node (S) is completely randomly generated to establish an extra tree; last the extra tree ensemble is built using the similarity method. Detailed information on ERT algorithm can be found in Geurts et al. (2006). Figure 4 illustrates the schematic of the enhanced STET model."

Figure 4. Schematic of the enhanced STET model developed in our study.

5. In Figure 7, what is surprising is that I see a good positive correlation pattern between R and RMSE. Generally, a good model performance is associated with a high R and a low RMSE against observations. Please check and clarify. **Response:** We have verified the numbers, which are correct. Mathematically speaking, R2 and RMSE are two independent measures of a correlation between two variables whose correlation depends on the slope of the regression between the two, higher for a regression slope closer to unity. Since the slope varies from site to site, they may not show the same spatial patterns. We have taken a closer look at the spatial patterns of these quantities and added the following text attempting to give a physical explanation (section 4.2.1 of the revised manuscript): "In general, high R2 with overall large RMSE but small MRE values are observed at the beginning and end of the year (in winter). This is because PM2.5 concentrations vary more and are always high due to the greater amount of pollutant emissions caused by heating or frequent dust storms. By contrast, lower R2 with overall small RMSE and large MRE values are observed in the middle of the year (in summer) because air pollution levels are lower."

Specific comments:

1. Line 48, the "evenly dispersed" is confusing, and is conflict with the "PM2.5 shows great spatial and temporal heterogeneities" in Line 80. **Response:** Corrected.

2. Line 175, better replace "differences" by variation.

Response: Corrected.

3. Line 227, typos: Figure 2 or Figure 3? **Response:** Corrected.

4. Line 247, what is definition for MAE and MRE? **Response:** We have provided definitions of these evaluation indicators in the revised manuscript.

5. Figure 9, typos: the year is 2018 or 2017? Also please add the season labels for each plot. **Response:** Corrected.

Reddington, C. L., Morgan, W. T., Darbyshire, E., Brito, J., Coe, H., Artaxo, P., Scott, C. E., Marsham, J., and Spracklen, D. V.: Biomass burning aerosol over the Amazon: analysis of aircraft, surface and satellite observations using a global aerosol model, Atmos. Chem. Phys., 19, 9125-9152, 10.5194/acp-19-9125-2019, 2019.

Sun, Y., Jiang, Q., Wang, Z., Fu, P., Li, J., Yang, T., and Yin, Y.: Investigation of the sources and evolution processes of severe haze pollution in Beijing in January 2013, Journal of Geophysical Research: Atmospheres, 119, 4380-4398, 2014.

Zheng, G., Duan, F., Su, H., Ma, Y., Cheng, Y., Zheng, B., Zhang, Q., Huang, T., Kimoto, T., and Chang, D.: Exploring the severe winter haze in Beijing: the impact of synoptic weather, regional transport and heterogeneous reactions, Atmos. Chem. Phys., 15, 2969-2983, 2015.

**Reviewer: 3**

I noticed that the same authors published a very similar paper in ES&T, https://pubs.acs.org/doi/10.1021/acs.est.9b03258. The only difference is between PM2.5 and PM1.0. However, the ACP paper needs originality.

**Response:** We would say that the two papers are similar but also differ in many regards that are grossly summarized as follows:

- (1) They deal with different pollution quantities: PM1 and PM2.5, whose emission sources, formation and transport mechanisms, and health impact are all different. As such, both the figures and text of the manuscripts differ considerably. Their ratio varies greatly, ranging from less than 0.5 to greater than 0.9 at both spatial and temporal scales, especially in heavily polluted regions due to different influential factors (Wei et al., 2019b). The two papers may thus be regarded as a series of companion studies that do not undermine their respective scientific originality. The reviewer is invited to compare them to see how different they are.
- (2) The estimation approaches used to derive PM1 and PM2.5 are similar but also differ in several aspects. While the same kind of machine learning method, namely, the space-time extra-trees (STET) model, is used for retrieving PM1 and PM2.5, there are numerous differences in their applications. For retrieving PM2.5, we have 1) used different input parameters by adding the aerosol precursor gases (SO2, CO, NOx, VOC, fine-size dust) from pollutant emission inventories; 2) corrected the satellite retrievals of AOD with reference to ground-based measurements; 3) modified the feature selection approach using the Gini index; and 4) improved the determination of spatiotemporal information. We have clearly described these differences in Section 3 as well as in the introduction of the revised manuscript.

Moreover, the manuscript has some fatal defects, (1) It does not work well with high pollution events, which is paid more attention.

**Response:** Like similar studies, ours suffers from a limitation of having relatively large errors under severely polluted conditions whose causes are further explained, per the reviewer's suggestion. This is a common problem reported in many previous studies. We have added the following text to the revised manuscript (Section 5.1): "We find that all traditional statistical regression models, and machine and deep approaches reported in previous studies underestimated PM2.5 concentrations under highly polluted conditions with poor regressions (i.e., slope < 0.9, and intercept > 6  $\mu$ g/m3) between measurements and retrievals of PM2.5 in China, a common problem. Potential causes are: 1) There are large estimation errors in AOD retrievals under severe pollution conditions in China (Wei et al., 2019c). This is further rooted to the fundamental limitations of satellite-based AOD retrievals, i.e., the non-linear to reflectance and the high sensitivity of the single-scattering albedo (Z. Li et al., 2009); 2) High AOD does not correspond to high PM2.5 concentrations because their ratio is highly variable over space and time, affected by both natural and human factors; 3)

The number of samples for high-pollution cases is small, hindering the ability to train the model."

It appears that all approaches suffer from this inherent limitation, which should thus not be regarded as a "fatal defect" of our study, more importantly, the comparison results suggest that our model can more accurately capture the high pollution events with a larger slope of 0.86 and a smaller intercept of 6.16  $\mu$ g/m3 with reference to other models reported from previous studies (Table 2).

(2) Such method seems falling into a dead cycle, the results were compared by the observations which were used to fit the parameters. I do not think it works with another independent database. Some similar comments were pointed by the other two reviewers.

**Response:** We do not think the method itself is a "dead cycle", but do make more efforts to enhance the validity and effectiveness of the validation approach. Three independent validation methods are applied, ensuring that the training and validation data are independent, as described in Section 3.5, copied below: "Different from our previous study, three independent validation methods are performed to verify the model's ability to estimate PM2.5 concentrations. The first independent validation method, i.e., the out-of-sample cross-validation (CV) approach, is performed by all data samples using the 10-fold CV procedure (Rodriguez et al., 2010). The data samples are divided into ten subsets randomly, and nine (one) of them are used as training (validation) data. This approach is repeated ten times, and error rates are averaged to obtain the final result. This is a common approach to evaluate the overall accuracy of a machine learning model, widely adopted in most satellite-derived PM studies (T. Li et al., 2017a, b; Ma et al., 2014, 2019; Xiao et al., 2017; He and Huang, 2018; Chen et al., 2019; Wei et al., 2019b; Xue et al., 2019; Yao et al., 2019).

The second independent validation method, i.e., out-of-station CV approach, is similar to the first one but performed using data from the monitoring stations to evaluate the spatial performance of the model. Data samples collected from different spatial points make up the training and testing data, and the relationship between spatial predictors and PM2.5 built from the training dataset is then estimated for each testing. The third independent validation approach tests the predictive power of the model. It is performed by applying the model built for one year to predict the PM2.5 concentrations for other years, then validating the results against the corresponding ground measurements. This approach ensures that the data samples for model training and validation are completely independent on both spatial and temporal scales."

**Improved 1-km-resolution PM2.5 estimates across China using theenhanced space-time extremely randomized trees**

Jing Wei1,2, Zhanqing Li2\*, Maureen Cribb2, Wei Huang3, Wenhao Xue1, Lin Sun4, Jianping Guo5,

- 5 Yiran Peng6, Jing Li7, Alexei Lyapustin8, Lei Liu9, Hao Wu1, Yimeng Song10
  - 1. State Key Laboratory of Remote Sensing Science, College of Global Change and Earth System Science, Beijing Normal University, Beijing, China
  - 2. Department of Atmospheric and Oceanic Science, Earth System Science Interdisciplinary Center, University of Maryland, College Park, MD, USA
  - 3. State Key Laboratory of Remote Sensing Science, Faculty of Geographical Science, Beijing Normal University, Beijing, China
  - 4. College of Geomatics, Shandong University of Science and Technology, Qingdao, China
  - 5. State Key Laboratory of Severe Weather, Chinese Academy of Meteorological Sciences, Beijing, China
- 15 6. Ministry of Education Key Laboratory for Earth System Modeling, Department of Earth System Science, Tsinghua University, Beijing, China
  - 7. Department of Atmospheric and Oceanic Sciences, School of Physics, Peking University, Beijing, China
  - 8. Laboratory for Atmospheres, NASA Goddard Space Flight Center, Greenbelt, Maryland, USA
  - 9. College of Earth and Environmental Sciences, Lanzhou University, Lanzhou, China
- 20 10. Department of Urban Planning and Design, Faculty of Architecture, The University of Hong Kong, Hong Kong

Correspondence to: Zhanqing Li (zli@atmos.umd.edu)

**25**

10

**Abstract**

Fine particulate matter with aerodynamic diameters  $\leq 2.5 \,\mu m \,(PM_{2.5}) \, shows has}$  adverse effects on human health and the atmospheric environment. Satellite derived aerosol products have been intensively adopted in estimating The estimation of surface PM2.5 concentrations, but has made intensive

30 use of satellite-derived aerosol products. However, most previous studies failed to monitor air pollution over small-scale areas, limited by the coarse spatial-resolution (3–50 km) and lowthe poor data-quality of aerosol optical depth (AOD) products. Therefore, a newHere, enhanced was the space-time extremely randomized trees (STET) model is developed that integrates by integrating updated spatiotemporal information and additional auxiliary data to improve PM2.5-estimates at boththe spatial

- 35 resolution and overall accuracy of PM2.5 estimates across China. To this end, the newly released MODIS MAIACModerate Resolution Imaging Spectroradiometer Multi-Angle Implementation of Atmospheric Correction AOD product, along with meteorological and other auxiliary data are inputs, topographical, land-use data and pollution emissions were input to the STET model. Daily, and daily 1km PM2.5 maps infor 2018 across mainland China arewere produced. The STET model
- 40 performsperformed well with a high out-of-sample (out-of-station) cross-validation coefficient of determination (R2) of 0.89 (0.88), a low root-mean-square error of 10.3533 (10.9793) μg/m3, a small mean absolute error of 6.7169 (7.1715) μg/m3, and a small mean relative error of 21.3728 % (23.77%), respectively. Particularly, it can well capture the69%). In particular, the model captured well PM2.5 concentrations at both regional and individual site scales. In addition, it posed a strong predictive power
- 45 (e.g., monthly  $R^2 = 0.80$ ) and can be used to predict the historical PM2.5 records. The North China Plain, the Sichuan Basin, and Xinjiang Province always are featured with high PM2.5 pollution levels, especially in winter. The STET model outperformsoutperformed most models presented in previous related studies.-, with a strong predictive power (e.g., monthly  $R^2 = 0.80$ ) which can be used to estimate historical PM2.5 records. More importantly, ourthis study provides a new approach to obtaintoward
- 50 obtaining high-spatial-resolution and high-quality PM2.5 estimates, which is important for air pollution studies overfocused on urban areas.

**1. Introduction**

Atmospheric particulate matter is a relatively stable suspension system withgeneral term describing all
kinds of solid and liquid particulate matter evenly dispersedparticles in the atmosphere. Fine particles are those particles in ambient air with aerodynamic diameters no more than 2.5 micrometers (PM2.5). Compared to coarser particles, PM2.5 areis rich in toxic and harmful substances and can directly enter the respiratory tract and alveoli of humans. Moreover, they have a long residence time and long transmission distance in the atmosphere (Aggarwal and Jain, 2015). Numerous studies have illustrated
that high PM2.5 concentrationconcentrations adversely affects affect human health (Peng et al., 2009;

Bartell et al., 2013; Chowdhury and Dey, 2016; Crippa et al., 2019; Song et al., 2019), severely impairs the atmospheric environment (Z. Li et al., 2017), and even significantly influences the cloud and precipitation systems bythrough aerosol radiative and microphysical effects (Koren et al., 2014; 2016; Seinfeld et al., 2016; Ceca et al., 2018). Silva et al. (2013) have shown that about 2.1 million people

- have died each year, resulting from the increasing PM2.5 concentrations around the world.
  Nowadays, air pollution is becoming more severe due to continuously increasing anthropogenic aerosols in developing countries, especially in China (He et al., 2011; Huang et al., 2014; M. Liu et al., 2017; Zhai et al., 2019). Fine particulate matters havematter has become the primary pollutant in urban environmentenvironments, garnering much scrutiny from the public (Han et al., 2014; L. Sun et al.,
- 2016; Wu et al., 2018). Therefore, the China Meteorological Administration began to establishestablished in 2004 a ground PM2.5 observation network to monitor the urban air quality as early as 2004 (Guo et al., 2009), followed by a denser network established by the Chinese Ministry of Environmental Protection sincein 2013. However, station-based monitoring is largely limited by the instruments and climatic conditions and cannot completely reflectcharacterize air pollution over large
- 75 areas. Satellite remote sensing technology has led to a variety of operational aerosol optical depth (AOD) products using mature aerosol retrieval algorithms (Levy et al., 2013; Lyapustin et al., 2018), which allows the leading to estimates of PM2.5 estimations at large scalescales due to their unanimously the positive relationshipsrelationship between AOD and PM2.5 
[revised manuscript text omitted]
 acrossin China, which can leadleading to PM2.5 maps with wider spatial-coverage PM2.5 maps. More importantly, the coverages. The number of valid data samples has also significantly increased by approximately 25–32% after combination than just using Terra or Aqua MAIAC products, which can improve the %, improving the model training ability.
  - 7

 $\begin{pmatrix} \tau_{T} = k_{1} \cdot \tau_{A} + b_{1} \\ \tau_{A} = k_{2} \cdot \tau_{T} + b_{2} \\ \tau_{C} = \operatorname{mean}(\tau_{T}, \tau_{A}) \end{pmatrix}$

205

where  $\tau_T$ ,  $\tau_A$ , and  $\tau_C$  denote the Terra, Aqua, and combined AODs.

In addition, dueDue to different spatial resolutions, all the 16-auxiliary variables arewere uniformly aggregated to a 1-km ( $\approx 0.01^{\circ} \times 0.01^{\circ}$ ) spatial resolution using the bilinear interpolation approach. After removing invalid or unrealistic values, there are 167,716 matched PM2.5-AOD samples and independent variables are-collected for 2018 in China.

**3.2 Potential effects of variables on PM2.5**

The potential relationships between all selected independent variables and PM2.5 measurements are first

- 210 investigated (Figure 3). AOD is highly positively related to PM2.5 measurements (R = 0.54), and all pollutant emissions, nighttime lights, and land use cover show positive effects on PM2.5. By contrast, all topographical variables and NDVI are negatively related to PM2.5. Moreover, except for ET (R = 0.24) and SP (R = 0.16), the other meteorological variables show opposite negative effects on PM2.5, especially for BLH (R = -0.22) and TEM (R = -0.17). In general, all the selected variables are
- 215 significantly correlated to PM2.5 measurements at the confidence level of 0.01 or 0.05 (two sides), so they are used as inputs to the STET model for preliminary training.

**3.3 Updated feature selection**

Due to the large number of independent variables considered, this will lead to the unavoidable over-

- fitting issuewill occur during the model training process. Therefore, the The model need bethus needs further adjusted adjustment by selecting more 
[revised manuscript text omitted]
- 325 These69%. In addition, compared to the sample-based validation, the out-of-station accuracy changes little, suggesting that the enhanced STET model can well estimate daily PM2.5 concentrations. Moreover, these results illustrate that spatiotemporal information areis crucial in improving the PM2.5-AOD relationships and should be carefully considered when introducing statistical regression models using remote sensing techniques.

330

335

**4.1.2 Regional-scale validation**

Figure 6 shows the sample-based 10-CV results of the enhanced STET model in PM2.5 daily estimates over eastern and western China (according to the widely used Heihe-Tengchong line), and four typical local-regions (Figure 1). The enhanced STET model performs differently over eastern and western China, mainly due to significant differences in land cover and climate conditions. There are 1289

uniformly distributed PM2.5 stations in eastern China, and 127,241 daily samples were collected. The STET-model performs wellin eastern China with a high sample-based CV-R2 equal to 0.90 and low estimation uncertainties, i.e., RMSE =  $9.7772 \ \mu g/m^3$ , MAE =  $6.4441 \ \mu g/m^3$ , and MRE = 19.2416%. By contrast, there are 294 unevenly and sparsely distributed PM2.5 stations in western China, thuswith

about three times fewer daily PM2.5 estimates were-collected. The model performance is overall poorer (e.g.,  $CV-R^2 = 0.86$ , and 85,  $RMSE = 11.9912.04 \mu g/m^3$ ,  $MAE = 7.56 \mu g/m^3$ ) than over eastern China. This is mainly contributed attributed to brighter surfaces (e.g., desert and bare land) with little vegetation coverage and harsh meteorological conditions over western China.

There were 33,733, 15,199, 6,209, and 6,470 daily samples collected from 233, 184, 95, and 107 uniformly distributed PM2.5 monitoring stations in the North China Plain (NCP), the Yangtze River

- 345 uniformly distributed PM2.5 monitoring stations in the North China Plain (NCP), the Yangtze River Delta (YRD), the Pearl River Delta (PRD)), and the Sichuan Basin (SCB), respectively. For former threeEstimated PM2.5 concentrations in the typical urban agglomerations where people closely concerned, the estimated PM2.5-concentrations of the NCP, YRD, and PRD are highly consistent with surface measurements (CV-R2 = 0.8986–0.92)), with overall low estimation uncertainties (i.e., RMSE =
- 350  $78-12 \ \mu\text{g/m}^3$ , MAE = 5-8  $\mu\text{g/m}^3$ , and MRE = 15-19%). In addition, the STETThe new model also performs well over the Sichuan Basin with an average CV-R2 value equal to 0.87 and comparable estimation uncertainties to North China Plain. In general those from the NCP. Overall, despite some differences in model performance, the enhanced STET model shows an overall good ability in estimating PM2.5 estimates concentrations at the regional scale.
- 355

**4.1.3 Site-scale validation**

National- and regional-scale aggregated evaluations mainly illustrate the overall performance of the STET-model in estimating PM2.5 estimates, howeverconcentrations. However, due to the inhomogeneity of PM2.5 monitoring stations, an additional validation for each monitoring station in China is performed
 (Figure 67). For statistical significance, plotted are only these monitoring stations with more than ten data samples are plotted. The daily. Daily PM2.5 estimations are estimates relate well related to surface measurements at most individual stations across China. The average sample-based CV-R2 is 0.84, and the CV-R2 values are highergreater than 0.8 at more than 73% of the monitoring stations, especially

forin eastern China. However, observed are relatively poorer performances (CV-R2 < 0.6) are observed

- at some scattered sites located in southwesternsouthwest and southeasternsoutheast China. In general, the STETnew model shows overall low estimation uncertainties at most sites with average RMSE and MAE values of 9.32 and 6.5  $\mu$ g/m3, especially forin southern China. Moreover, the average RMSE and MAE values are < 10  $\mu$ g/m3 at more than 68% and 93~94% of the monitoring stations across China.in China have mean RMSE and MAE values less than 15  $\mu$ g/m3 and 10  $\mu$ g/m3, respectively. Note that
- 370 these stations showhave larger RMSE values (> 10 μg/m3) in central China, mainly due to the high pollutedpollution levels. In addition, the The average MRE value in China is 20.888%, and most stations (> 86%)% of them) have low MRE values <less than 30% in PM2.5 estimations in China,%, especially for those states located in eastern and southern China.

**375 **4.2 Performance at the temporal scale**

**4.2.1 Daily-scale validation**

Figure 78 shows the STET model performance from all available monitoring stations in China as a function of the day of yearDOY. The number of data samples in one day ranges from 54 to 1155, with an average of 466 in 2018. In general, the STETnew model shows great performanceperforms well
(average CV-R2 = 0.76) at 77) on most days in the year, and more than 7677% of thethese days have CV-R2 values greater than 0.7. Two main uncertainty metrics, i.e., RMSE and MAE, show similar temporal variations during the year, first decreasing until around day 250, then gradually increasing. In general, approximately equal Approximately 91% and 92% of the days have low RMSE and MAE values of less than 15 and 10 μg/m3, respectively, over the year. Large estimation uncertainties always
occur at the beginning and end of the year mainly due to intense human activities and harsh natural

- environment. Furthermore, MRE is relatively stable, ranging from 13% to 5249% with an average value of 23.292%, and more than 87% of the days yield lowhave MRE values of less than 30% in China. These results illustrate that the STET model show great performance in capturingIn general, high R2 with overall large RMSE but small MRE values are observed at the beginning and end of the year (in
- 390 winter). This is because PM2.5 concentrations on most days of the year.vary more and are always high due to the greater amount of pollutant emissions caused by heating or frequent dust storms. By contrast,

lower R2 with overall small RMSE and large MRE values are observed in the middle of the year (in summer) because air pollution levels are lower. Nevertheless, these results illustrate that the enhanced STET model captures well PM2.5 concentrations on most days of the year.

395

**4.2.2 Seasonal-scale validation**

Figure 9 shows sample-based cross-validationCV results for PM2.5 daily estimates divided by four seasonsaccording to the season in 2018 acrossin China. The resultsResults suggest that there are obviousclear differences in model performance at the seasonal level. The the number of valid data

- 400 samples because of the long-term snow/ice cover in winter and more frequent clouds in summer, resulting in an overall smaller number of samples than in the other two seasons. The enhanced STET model performs best in autumn with the highest CV-R2 value of 0.90 and the strongest regression line (i.e., slope = 0.88, and intercept =  $4.8885 \ \mu g/m^3$ ). The averageMean RMSE, MAE3 and MRE values in autumn are 9.018.97  $\mu g/m^3$ , 5.8784  $\mu g/m^3$ , and 21.10-02%, respectively. By contrast, the STET new
- 405 model performs the worst in summer with the lowest  $CV-R^2$  of 0.7679 and smallesta less steep slope of 0.747.37, indicating obvious clear underestimations. However, summer shows experiences the least amount of air pollution with most daily PM2.5 values  $< 8050 \mu g/m^3$ , leading to smallest estimation uncertainties. The main reason is that the meteorological conditions in place in summer accelerated the diffusion of pollutants but complicated the PM2.5-AOD relationships. The airthe smallest RMSE and
- 410 MAE values but the largest MRE values. Air quality is about two or three times worse in spring and winter than in winter with wider PM2.5 ranges and larger standard deviations. Moreover, the STETThe model showsperformance in these seasons is similar performances in these two seasonal, with almost equal CV-R2 and slope values, as well as and close estimation uncertainties. The differences in model performance among the seasons are mainly attributed to seasonal variations in natural conditions and
- 415 human activities. Meteorological conditions in summer favor the diffusion of pollutants but complicate the PM2.5-AOD relationship (Su et al., 2018, 2020), whereas direct emissions of pollutants are greater in winter, resulting in severe air pollution.

**4.2.3 Synthetic-scale validation**

- 420 Synthetized PM2.5 retrievals are validated against PM2.5 surface observations by calculating the effective values from the same number of valid days at monthly, seasonal, and annual time scales (Figure 10). Monthly PM2.5 estimates and ground measurements (N = 12,410) are highly correlated (R2 = 0.93), with a steep slope of 0.91. Mean RMSE, MAE, and MRE values are 5.63 µg/m3, 4.08 µg/m3, and 11.59%, respectively. Seasonal mean PM2.5 estimates (N = 5,231) have a good accuracy (i.e., R2 = 0.93, RMSE =
- 425 5.00  $\mu$ g/m3, MAE = 3.69  $\mu$ g/m3, and MRE = 10.31%). Annual mean PM2.5 estimates (N = 1,462) agree well with ground measurements (R = 0.91), with small uncertainties (i.e., RMSE = 4.11  $\mu$ g/m3, MAE = 3.12  $\mu$ g/m3, and MPE = 8.58%). This illustrates that the synthetic dataset can more accurately reflect the spatiotemporal PM2.5 loadings and variations across China.

**430 2.74.3 Predicted PM2.5 maps across China**

The monthly Monthly PM2.5 maps are thus synthesized and averaged from at least 20% of available daily PM2.5 estimates for each grid in a month-in 2018-, and annual PM2.5 maps are generated from monthly PM2.5 maps if there are more than eight available values for each grid across China (Hsu et al., 2012; Wei et al., 2019f). The spatial coverage of monthly PM2.5 maps varies from 73% to 92%, with an

- 435 average of 83% across mainland China. The highest (lowest) spatialmaximum coverage occurs around October (occurs in April, and the minimum coverage occurs in January) of the year. Similarly, the. The monthly mean PM2.5 values vary conversely from 21.224.4 μg/m3 to 45.142.9 μg/m3 with, where the highest (lowest) PM2.5 concentration occurring around Marchis observed in December (August) of the year.
- 440 The satellite-derived 1-km-resolution PM2.5 map in 2018 covers almost the full scene (spatial coverage = 99%) across mainland China (Figure 11a) and is highly consistent in spatial patterns are similar between the STET-derived 1-km PM2.5 map and calculated in-pattern with the corresponding in situ measurements (Figure 11b). The average PM2.5 concentration is 32.7±13.6 µg/m3 in 2018 across mainland China. In general, the most severe PM2.5 pollution occurs in the Taklamakan Deseret, where
- 445 most areas expose are exposed to high PM2.5 concentrations of > 80  $\mu$ g/m3. There are also high-polluted pollution levels over the North China Plain, Sichuan Basin, and Yangtze River DeltaNCP, the SB, and the YRD, with annual mean PM2.5 values of 46.8±11.8, 38.37±10.35, 39.8±9.9, and 37.6±938.4±8.3

μg/m3, respectively. These mainly contributed to, arising from intensive human activities, and special topographic and meteorological conditions. By contrast, the annual mean PM2.5 loadings are loading is

- 450 overall low inover the rest areas of China, e.g., the PRD (33.4±3.9 μg/m3). However, there may be poor representativeness for these areas overin western China with few ground monitoring stations. In general, we have to say that the PM2.5 pollution has been significantly reduced in 2018 across China due to the effective emission control measures implemented by the Chinese government (Fang et al., 2019; Ma et al., 2019). However, more More than 3034% of mainland China-still experienced high PM2.5 levels in
- 2018 exceeding the international and national recommended air quality level (PM2.5 > 35 μg/m3).
  Figure 12 shows seasonal mean PM2.5 maps, which are averaged from the available monthly values for each grid, in 2018 across China. The average PM2.5 concentration (spatial coverage) is 37.2±20.7 μg/m3 (~ 96%), 25.5±12.1 μg/m3 (~ 92%), 29.5±11.5 μg/m3 (~ 97%), and 41.3±15.4 μg/m3 (~ 88%) for spring, summer, autumn, and winter, respectively. There are noticeable spatial differences in PM2.5
- 460 distributions on the seasonal scale. In winter and spring, more than 7749% and 6642% of mainland China exposing the were exposed to high PM2.5 levels > of 30 µg/m3, yielding poorer airresulting in poor quality. By contrast, PM2.5 pollution is slighter lower in summer and autumn, with more than 9190% and 8174% of mainland China, respectively, experiencing low PM2.5 levels below the acceptable air quality level. Note that in spring, PM2.5 concentrations are particularly high in Xinjiang province due to

465 frequent sand and dust episodes in 2018.

**5. Discussion**

**5.1 Model accuracy**

There is an increasing number of studies on estimating PM2.5 using satellite AOD products from local to national scales across China. However, limited by the operational satellite aerosol products, PM2.5 can only be estimated at coarse spatial resolutions of approximately 6–10 km (Fang et al., 2016; T. Li et al., 2017b; Yu et al., 2017; Chen et al., 2018; Ma et al., 2019; Yao et al., 2019). Recently, with the release of MODIS 3-km DT aerosol products, the PM2.5 estimates can be improved to a 3-km spatial resolution across China (You et al., 2016; T. Li et al., 2017a; He & and Huang, 2018; Chen et al., 2019; Xue et al.,

475 2019). Therefore, in our This study, improves the spatial resolution of PM2.5 estimates has been

significantly improved by 3–10 timesacross mainland China to 1 km based on the newly released highquality MAIAC products across mainland China.

ForRegarding model performance, our newly developed STET model shows much higher accuracy is more accurate with higher CV-R2 values, and smaller RMSE and MAE values than the those from

- 480 statistical regression models (Table 2), e.g., the timely structure adaptive model (TSAM5; Fang et al., 2016) model, the Gaussian model (Yu et al., 2017), the Generalized Additive Model (GAM5; Chen et al., 2018) model, and the GWR model (Ma et al., 2014; You et al., 2016), and the geographically and temporally weighted regression model (GTWR-model (; He and Huang, 2018). The enhanced STET model can also outperform most machine learning (ML) and deep learning approaches including the
- 485 RFGaussian model (Yu et al., 2017), the Random Forest model (Chen et al., 2018; Wei et al., 2019e), the XGBoost model (Chen et al., 2019), the Geo-BPNN, GRNN and deep brief network (DBN) models (T. Li et al., 2017a, 2017bb), and some optical combined models, e.g., the Daily-GWR model (D-GWR) model (; He and Huang, 2018), the two-stage model (He and Huang, 2018; Ma et al., 2019; Yao et al., 2019), and the ML + GAM model (Xue et al., 2019).
- 490 We find that all traditional statistical regression models, and machine and deep approaches reported in previous studies underestimated PM2.5 concentrations under highly polluted conditions with poor regressions (i.e., slope < 0.9 and intercept > 6  $\mu$ g/m3) between measurements and retrievals of PM2.5 in China, a common problem. Potential causes are: 1) There are large estimation errors in AOD retrievals under severe pollution conditions in China (Wei et al., 2019c). This is further rooted to the fundamental
- 495 limitations of satellite-based AOD retrievals, i.e., the non-linear to reflectance and the high sensitivity of the single-scattering albedo (Z. Li et al., 2009); 2) High AOD does not correspond to high PM2.5 concentrations because their ratio is highly variable over space and time, affected by both natural and human factors; 3) The number of samples for high-pollution cases is small, hindering the ability to train the model. Therefore, our model also tends to underestimate PM2.5 concentrations on highly polluted
- 500 days ( $PM_{2.5} > 150 \ \mu g/m^3$ ), however, it can more accurately capture the high pollution events with a stronger slope of 0.86 and a smaller intercept of 6.16  $\mu g/m^3$  with reference to other models reported from previous studies (Table 2).

Furthermore, compared with daily PM1 estimates using the STET model in our previous study (CV- $R^2$  = 0.76 and slope = 0.70; Wei et al., 2019b), the overall accuracy of daily PM2.5 estimates using the

505 enhanced STET model has improved significantly with a much higher CV-R2 of 0.89 and a steeper slope of 0.86, based on data from 2018 in China. Continuous improvements of the model can further improve the determination of the relationship between fine particulate matter and AOD so as to improve the model performance. More data samples may also help improve the training ability of the model.

**510 **5.2** Predictive power**

To test the predictive power in PM2.5 concentrations of the enhanced STET model, the model built for the year of 2018 was used to predict daily PM2.5 concentrations in 2017, validated against the ground measurements from 2017. Results suggest that our new model can correctly capture more than 65% of the historical daily PM2.5 concentrations (N = 177,616). Monthly (N = 12,408), seasonal (N = 5,227),

- 515 and annual (N = 1,461) mean PM2.5 predictions across China. The comparison results are highly correlated with surface observations with R2 values of 0.80, 0.81, and 0.82, respectively, having overall small estimation uncertainties (i.e., RMSE < 12  $\mu$ g/m3, MAE < 9  $\mu$ g/m3, and MRE < 26  $\mu$ g/m3). There are only a handful of studies examining the predictive powers of models estimating PM2.5 concentrations in China. Comparisons show that <del>our</del>the enhanced STET model is superior to those
- 520 results-reported byin previous studies, i.e., the two-stage model (Ma et al., 2019), the GTWR model (He and Huang, 2018), the ML + GAM model (Xue et al., 2019), and the STRFspace-time RF model (Wei et al., 2019e). The enhanced STET model has a strong predictive power and can be used to estimate historical PM2.5 concentrations in China.

**525 **3.6.**Summary and conclusions**

With the increase in air pollution over recent years, abundant studies on estimating PM2.5 have been performed using satellite remote sensing. However, most of the PM2.5 estimates are reported at spatial resolutions of 3–10 km, which is inadequate for monitoring air quality atin urban areas. The Traditional models also limit the accuracy of PM2.5 estimates is also limited by traditional models. Therefore. Here,

530 we try to generate present spatially continuous high-quality PM2.5 maps at a 1-km higher spatial

resolution across China. For this, a new space-time extremely randomized trees (an enhanced STET) approach is model was developed to minimize the spatiotemporal heterogeneities in PM2.5 and improve the overall estimate accuracy of ground-level PM2.5 concentrations.

- Our results suggest that the enhanced STET model <del>shows great performance in estimatingestimates well</del> 535 daily PM2.5 concentrations at the national scale with a relatively high sample-based cross-validation coefficient of 0.89, low RMSE of 10.35 µg/m3, MAE of 6.71 µg/m3, and MRE of 21.37% at the national scale.%. Comparisons illustrate that spatiotemporal information is of great importanceimportant and should be carefully considered during model development. The enhanced STET model <del>shows better</del> <del>performanceestimates PM2.5 concentrations well at most monitoring stations and individual days in the</del>
- 540 year. The North China Plain and the Sichuan Basin regions, under the influence of intense human activities and poor dispersion conditions, have high PM2.5 loadings. Moreover, the The enhanced STET model can outperform most models presented in previous related studies in terms of spatial resolution, model accuracya and predictive power. This study suggests that the 1-km-resolution PM2.5 dataset will be of great importance useful in future atmospheric pollution studies focused on medium- or small-scale
- 545 areas. In addition, the The enhanced STET model willmay be applied in the future to produce the historical PM2.5 dataset acrossdatasets for China in our future studies since because the 
[revised manuscript text omitted]

|                                   | RH                    | Relative humidity                                         | %                     | 0.125°×0.125°         | 3-hour                 | <del>reanalysis</del>
<del>product</del> |  |
|                                   | TEM                   | 2-m air temperature                                       | Κ                     |                       | 6-hour                 |                                             |  |
|                                   | SP                    | Surface pressure                                          | hPa                   |                       | 6-hour                 |                                             |  |
|                                   | WS                    | 10-m wind speed                                           | m/s                   |                       | 6-hour                 |                                             |  |
|                                   | WD                    | 10-m wind direction                                       | m/s                   |                       | 6-hour                 |                                             |  |
| Land coveruse                     | NDVI                  | NDVI                                                      | -                     | 500 m × 500 m         | Monthly                | MOD13A3                                     |  |
|                                   | LUC                   | Land use cover                                            | -                     | 500 m × 500 m         | Annually               | MCD12Q1                                     |  |
| Topography                        | DEM                   | DEM                                                       | m                     |                       |                        |                                             |  |
|                                   | Relief                | Surface relief                                            | m
00 m × 00 m      |                       |                        | SDTM                                        |  |
|                                   | Aspect                | Surface aspect                                            | degree                | 90 m × 90 m           | -                      | SKTM                                        |  |
|                                   | Slope                 | Surface slope                                             | degree                |                       |                        |                                             |  |
| Emission                   | SO2            | Sulfur dioxide                                            |                       |                       |                        |                                             |  |
|                                   | NOx | Nitrogen oxide                                            |                       |                       | Monthly         | MEIC                                        |  |
|                                   | CO             | Carbon monoxide                                           | Mo/orid               | 0 25°×0 25°           |                        |                                             |  |
|                                   | VOC            | Volatile organic                                          | 1715/ 5114     | 0.20 0.20             |                        |                                             |  |
|                                   |                       | compounds                                          |                       |                       |                        |                                             |  |
|                                   | Dust           | Fine-sized dust                                           |                       |                       |                        |                                             |  |
| Population                        | NTL                   | Night lights                                              | W/cm 2 /sr | 500 m × 500 m         | Monthly                | VIIRS                                       |  |

Table 1. Summary of the data sources used in this study.

[revised manuscript text omitted]